# A high affinity RIM-binding protein/Aplip1 interaction prevents the formation of ectopic axonal active zones

Matthias Siebert[1†], Mathias A Böhme[1,2†], Jan H Driller[3], Husam Babikir[1], Malou M Mampell[1], Ulises Rey[1,4], Niraja Ramesh[1], Tanja Matkovic[1], Nicole Holton[3], Suneel Reddy-Alla[1], Fabian Göttfert[5], Dirk Kamin[5], Christine Quentin[1], Susan Klinedinst[6], Till FM Andlauer[1], Stefan W Hell[5], Catherine A Collins[6], Markus C Wahl[3], Bernhard Loll[3], Stephan J Sigrist[1,2]*

[1]Institute for Biology/Genetics, Freie Universität Berlin, Berlin, Germany; [2]NeuroCure, Charité-Universitätsmedizin Berlin, Berlin, Germany; [3]Institute of Chemistry and Biochemisty/Structural Biochemistry, Freie Universität Berlin, Berlin, Germany; [4]Department of Theory and Bio-Systems, Max Planck Institute of Colloids and Interfaces, Potsdam, Germany; [5]Department of Nanobiophotonics, Max Planck Institute for Biophysical Chemistry, Göttingen, Germany; [6]Department of Molecular Cellular and Developmental Biology, University of Michigan, Ann Arbor, United States

**Abstract** Synaptic vesicles (SVs) fuse at active zones (AZs) covered by a protein scaffold, at *Drosophila* synapses comprised of ELKS family member Bruchpilot (BRP) and RIM-binding protein (RBP). We here demonstrate axonal co-transport of BRP and RBP using intravital live imaging, with both proteins co-accumulating in axonal aggregates of several transport mutants. RBP, via its C-terminal Src-homology 3 (SH3) domains, binds Aplip1/JIP1, a transport adaptor involved in kinesin-dependent SV transport. We show in atomic detail that RBP C-terminal SH3 domains bind a proline-rich (PxxP) motif of Aplip1/JIP1 with submicromolar affinity. Pointmutating this PxxP motif provoked formation of ectopic AZ-like structures at axonal membranes. Direct interactions between AZ proteins and transport adaptors seem to provide complex avidity and shield synaptic interaction surfaces of pre-assembled scaffold protein transport complexes, thus, favouring physiological synaptic AZ assembly over premature assembly at axonal membranes.

*For correspondence: stephan.sigrist@fu-berlin.de

†These authors contributed equally to this work

Competing interests: The authors declare that no competing interests exist.

## Introduction

The primary function of the presynaptic active zone (AZ) is to regulate the release of neurotransmitter-filled synaptic vesicles (SVs) in response to action potentials entering the synaptic bouton (*Südhof, 2012*). Before AZ scaffold components (e.g., ELKS family protein Bruchpilot: BRP, Rab3-interacting molecule (RIM)-binding protein: RBP) are integrated into synapses, however, they have to be transported down the often very long axons. AZ scaffold proteins are characterized by strings of interaction motifs (particularly coiled coil motifs) contributing to the avidity and tenacity of synaptic scaffolds (*Tsuriel et al., 2009*). Therefore they might be considered as 'sticky cargos' whose association status has to be precisely controlled during transport. Long-range axonal transport is conducted along polarised microtubules, using kinesin-family motor proteins for anterograde and dyneins for retrograde transport (reviewed in *Maeder et al., 2014*). Kinesin-1 family motor kinesin heavy chain (KHC, also known as KIF5; *Sato-Yoshitake et al., 1992*; *Hurd and Saxton, 1996*; *Takamori et al., 2006*) and Unc-104/Imac/KIF1 (*Hall and Hedgecock, 1991*; *Pack-Chung et al., 2007*) have been

**eLife digest** To pass on information, the neurons that make up the nervous system connect at structures known as synapses. Chemical messengers called neurotransmitters are released from one neuron, and travel across the synapse to trigger a response in the neighbouring cell. The formation of new synapses plays an important role in learning and memory, but many aspects of this process are not well understood.

In a specific region of the synapse called the active zone, a scaffold of proteins helps to release the neurotransmitters. These proteins are made in the cell body of the neuron, and are then transported to the end of the long, thin axons that protrude from the cell body. This presents a challenge for the cell, because the components of the active zone scaffold must be correctly targeted to the synapse at the end of the axon, ensuring the active zone scaffold assembles only at its proper location.

Siebert, Böhme et al. studied how some of the proteins that are found in the active zone scaffold of the fruit fly *Drosophila* are transported along axons. Labelling the proteins with fluorescent markers allowed their movement to be examined under a microscope in living *Drosophila* larvae. The results showed that two of the proteins—known as BRP and RBP—are transported along the axons together. Further investigation revealed that a transport adaptor protein called Aplip1, which binds to RBP, is required for this movement. Siebert, Böhme et al. established the structure of the part of RBP where this interaction occurs, and found that mutating this region causes premature active zone scaffold assembly in the axonal part of the neuron. The interaction between RBP and Aplip1 is very strong, and this helps to prevent the scaffold assembling before it has reached the correct part of the neuron. Exactly how the transport adaptor and active zone protein are separated once they reach their final destination (the synapse) remains to be discovered.

implicated in the transport of SVs, in conjunction with regulators of this process, such as Syd-1 (*Hallam et al., 2002*), Syd-2/Liprin-α (*Serra-Pagès et al., 1998*; *Zhen and Jin, 1999*; *Miller et al., 2005*; *Stryker and Johnson, 2007*; *Wagner et al., 2009*), RSY-1 (*Patel and Shen, 2009*), or ARL-8 (*Klassen et al., 2010*; *Wu et al., 2013*). In *Caenorhabditis elegans*, SV and AZ scaffold proteins exhibit extensive co-transport and undergo frequent pauses, with immobile phases promoting cargo dissociation and assembly (*Wu et al., 2013*). Long axons, typical for *Drosophila* or mammals, pose high demands for the 'processivity' of axonal AZ scaffold component transport. The molecular mechanisms, which provide this processivity and thus block premature assembly processes remain speculative, but might also be relevant in the context of axonal transport deficits of neurodegenerative scenarios (*Millecamps and Julien, 2013*). In addition, we know little concerning the composition of cargos destined for synaptic AZs.

The electron-dense AZ cytomatrix (T-bar) at the *Drosophila* neuromuscular junction (NMJ) is among others composed of oligomers of BRP and RBP (*Kittel et al., 2006*; *Fouquet et al., 2009*; *Liu et al., 2011a*; *Ehmann et al., 2014*). We report here that BRP and RBP, but no other tested AZ components, are co-transported in discrete transport complexes along the axon. Via a screen for RBP interaction partners, we identified the APP-like protein interacting protein 1 (Aplip1), an adaptor protein previously implicated in SV transport. Further analysis by X-ray crystallography and calorimetry showed that the second and third Src homology 3 (SH3) domain of RBP bind a specific N-terminal proline-rich (PxxP) motif of Aplip1/JIP1 with more than 10-fold higher affinity than RBP binds its synaptic ligands ($Ca^{2+}$channels/RIM) by their cognate PxxP motifs. The integrity of this motif was essential to protect axons from forming ectopic axonal synapses, which were observed in *aplip1* mutant axons by electron microscopy (EM) and super-resolution light microscopy.

In summary, we characterize a mechanism of axonal AZ protein transport through a high affinity interaction between preassembled, stoichiometric scaffold protein complexes and the transport adaptor Aplip1. This high affinity interaction is needed to allow for effective axonal transport and to protect from premature AZ assembly processes.

## Results

The molecular basis of how axonal protein transport is coupled to AZ assembly remains largely unexplored. We hypothesized that BRP might be co-transported with further AZ scaffold proteins, as

transport of preformed complexes of AZ material has been suggested previously (*Zhai et al., 2001*; *Shapira et al., 2003*; *Maas et al., 2012*).

## RBP co-clusters with BRP in axonal aggregates of SR kinase mutants

Firstly, we chose a previously characterized mutant of a serine–arginine (SR) protein kinase at location 79D (srpk79D). The SRPK79D protein is a member of the serine–arginine protein kinase family previously shown to be involved in mRNA splicing and processing (*Wang et al., 1998*). Mutants of *srpk79D* form dramatic BRP aggregates in the axoplasm, while its endogenous substrates remain elusive (*Johnson et al., 2009*; *Nieratschker et al., 2009*). The axonal aggregations here served as a sensitive background to screen for proteins that co-accumulate together with BRP in the axon, and therefore indicate a joint transport mechanism.

In order to visualise the aggregates forming within axons of *srpk79D* mutant larvae, we stained with antibodies (Abs) directed against the BRP C- and N-terminus (*Figure 1A*, as control), and further probed for the presence of additional AZ proteins, such as Liprin-α (*Figure 1B*) and Syd-1 (*Figure 1C*), which interact with BRP at the AZ (*Owald et al., 2010*, *2012*) and the small GTPase Rab3 that was previously shown to regulate the distribution of presynaptic components at AZs (*Figure 1D*; *Graf et al., 2009*). However, none of these AZ proteins showed co-accumulation with BRP in the aggregates (B as also described in *Johnson et al., 2009*). Staining with anti-RBP Abs (*Liu et al., 2011a*), by contrast, revealed strong co-localization of BRP and RBP in the axonal aggregates (*Figure 1E*). Quantification of BRP and RBP co-localization in two different *srpk79D* mutant null alleles (atc from *Johnson et al., 2009*; vn from *Nieratschker et al., 2009*) confirmed the impression that the axonal RBP/BRP signals were of identical size (*Figure 1F*; mean area of axonal spots, BRP$^{C-term}$ 0.3797 ± 0.03694 μm$^2$ in srpk79D$^{ATC}$, 0.3259 ± 0.02212 μm$^2$ in srpk79D$^{vn}$; RBP$^{C-term}$ 0.3892 ± 0.02097 μm$^2$ in srpk79D$^{ATC}$, 0.3696 ± 0.01645 μm$^2$ in srpk79D$^{vn}$; n = 8 nerves; mean ± SEM), and that BRP and RBP nearly always co-localized in these aggregates (*Figure 1G*; BRP$^{C-term}$ co-localizing with RBP$^{C-term}$ 93.26% ± 2.172 in srpk79D$^{ATC}$, 95.85% ± 1.302 in srpk79D$^{vn}$; RBP$^{C-term}$ co-localizing with BRP$^{C-term}$ 95.7% ± 0.9713 in srpk79D$^{ATC}$, 94.24% ± 1.162 in srpk79D$^{vn}$; n = 8 nerves; mean ± SEM).

Thus, RBP was the only AZ protein that robustly co-accumulates with BRP in *srpk79D* mutant axonal aggregates. To further explore the distribution of BRP and RBP in these aggregates we used stimulated emission depletion (STED) light microscopy at a resolution of about 50 nm (*Hell, 2007*). Two-colour STED microscopy revealed a tight and stoichiometric association of BRP and RBP in the floating axonal aggregates of *srpk79D* mutants (*Figure 1H*), reminiscent of EM images showing T-bar super assemblies in these axons (*Figure 1H*; *Johnson et al., 2009*; *Nieratschker et al., 2009*). In fact, the relative distribution of RBP vs BRP$^{C-term}$ was very reminiscent of the organisation at mature, synaptic AZs (*Liu et al., 2011a*). The tight association of BRP and RBP in these ectopic aggregates further suggested a co-transport of both AZ components. Indeed, we could identify axonal BRP spots co-positive for RBP (*Figure 1I*, arrows) in wild type (WT) larvae as well. Compared to *srpk79D* mutant axons, WT BRP/RBP co-positive aggregates were present at a lower frequency and displayed a ∼ four times smaller average diameter in control axons (*Figure 1F*; mean area of axonal spots, BRP$^{C-term}$ 0.06895 ± 0.01 μm$^2$ in WT; RBP$^{C-term}$ spots: 0.09184 ± 0.0133 in WT; n = 8 nerves; mean ± SEM).

## BRP and RBP are co-transported in axons together with Aplip1

We observed active anterograde and retrograde transport of the BRP (GFP-labelled)/RBP (cherry-labelled) co-positive spots when using intravital imaging of axons of intact larvae (*Rasse et al., 2005*) (*Figure 2A*; *Video 1*). Thus, as our data strongly suggested that BRP and RBP are co-transported, we searched for adaptor proteins coupling them to axonal motors.

RBP, via its second and third SH3 domain, is known to bind synaptic ligands such as Ca$^{2+}$ channels and RIM (*Liu et al., 2011a*). Both the SH3 domains and the cognate PxxP motifs of the synaptic ligands are highly conserved between mammals and *Drosophila* (*Liu et al., 2011a*; *Südhof, 2012*; *Davydova et al., 2014*). However, in order to identify novel RBP interaction partners which might be relevant in the context of axonal transport, we performed a large-scale yeast two-hybrid (Y2H) screen using a construct consisting of the second and third SH3 domains of *Drosophila* RBP as bait (also shown in Figure 3A). As expected, several clones representing RIM and the Ca$^{2+}$ channel α1-subunit Cacophony (Cac) were isolated (not shown). In addition, the screen recovered 14 independent fragments of Aplip1, including a full length cDNA clone (*Figure 2B*). Aplip1 is the *Drosophila* homolog

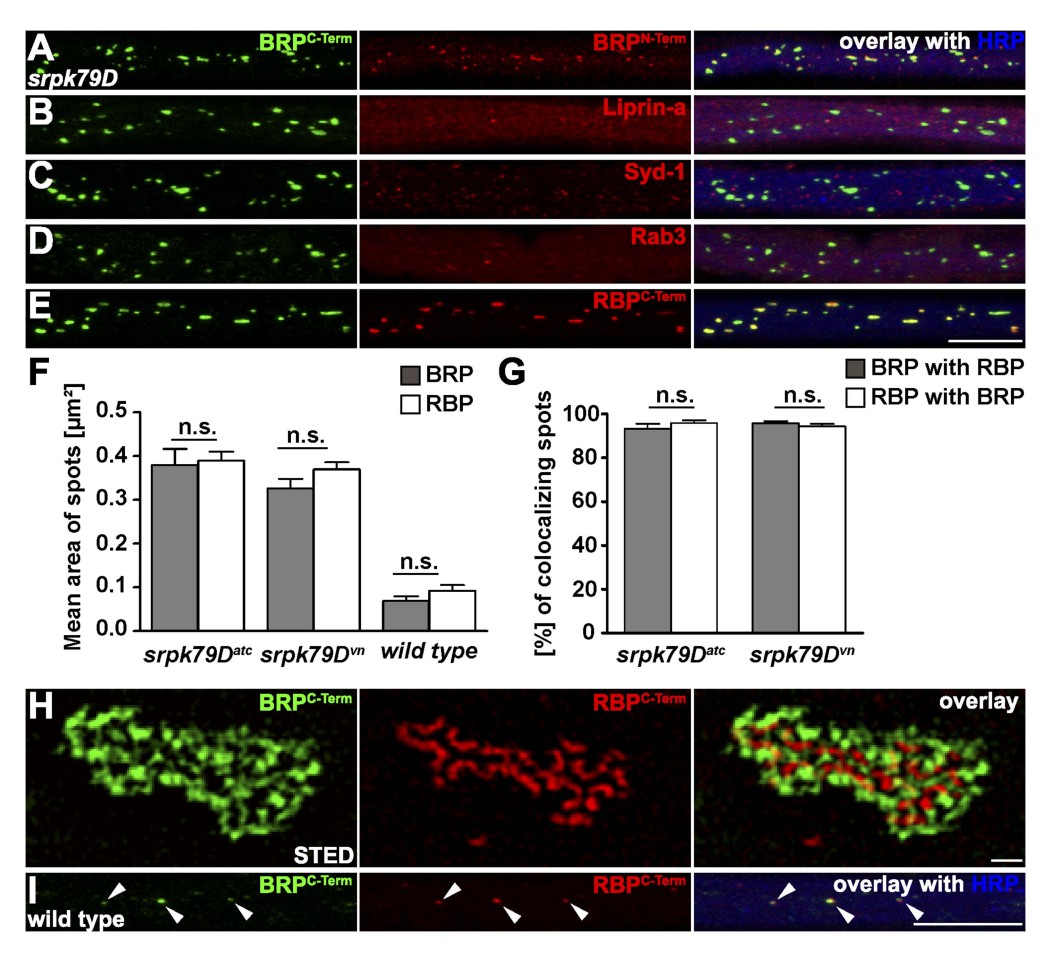

**Figure 1.** Co-accumulation of Bruchpilot (BRP) and RIM-binding protein (RBP) in *srpk79D* axonal aggregates.
(**A–E**, **I**) Nerve bundles of segments A1–A3 from third instar larvae of the genotypes indicated labeled with the
antibodies (Abs) indicated. (**A–E**, **H**) BRP accumulated in axonal aggregates of *srpk79D* mutants. (**B–D**) Liprin-α
(**B**), Syd-1 (**C**), and Rab3 (**D**), did not co-localize with axonal BRP spots. (**E**) By contrast, RBP invariably co-localized with BRP
in these axonal aggregates. (**F**) Quantification of mean area of axonal BRP and RBP spots in wild type (WT) and *srpk79D*
mutants. BRP$^{C-term}$ spots: $0.3797 \pm 0.03694$ µm$^2$ in *srpk79D$^{ATC}$*, $0.3259 \pm 0.02212$ µm$^2$ in *srpk79D$^{vn}$*, $0.06895 \pm 0.01$ µm$^2$
in WT; RBP$^{C-term}$ spots: $0.3892 \pm 0.02097$ µm$^2$ in *srpk79D$^{ATC}$*, $0.3696 \pm 0.01645$ µm$^2$ in *srpk79D$^{vn}$*, $0.09184 \pm 0.0133$ in WT;
n = 8 nerves each; all panels show mean values and errors bars representing SEM; ns, not significant, $p > 0.05$,
Mann–Whitney U test. (**G**) Quantification for BRP co-localization with RBP and vice versa in *srpk79D* mutants. BRP$^{C-term}$ co-
localizing with RBP$^{C-term}$: $93.26\% \pm 2.172$ in *srpk79D$^{ATC}$*, $95.85\% \pm 1.302$ in *srpk79D$^{vn}$*; RBP$^{C-term}$ co-localizing with BRP$^{C-term}$:
$95.7\% \pm 0.9713$ in *srpk79D$^{ATC}$*, $94.24\% \pm 1.162$ in *srpk79D$^{vn}$*; n = 8 nerves each; all panels show mean values and errors
bars representing SEM; ns, not significant, $p > 0.05$, Mann–Whitney U test. (**H**) Two-colour stimulated emission depletion
(STED) images of axonal aggregates in *srpk79D* mutants revealed that RBP$^{C-Term}$ label localized to the inside of the axonal
aggregates and was surrounded by BRP$^{C-Term}$ label. (**I**) BRP and RBP also co-localized in axonal spots of WT animals (arrow
heads show co-localization of BRP and RBP in the axon). Scale bars: (**A–E**, **I**) 10 µm; (**H**) 200 nm.

of c-Jun N-terminal kinase (JNK)-interacting protein 1 (JIP1), a scaffolding protein that has been
shown to bind kinesin light chain (KLC; *Verhey et al., 2001*), Alzheimer's amyloid precursor protein
(APP; *Taru et al., 2002*), JNK pathway kinases (*Horiuchi et al., 2005*, *2007*) and the autophagosome
adaptor LC3 (*Fu et al., 2014*). If Aplip1 was mediating the axonal transport of RBP, moving spots co-
positive for both RBP and Aplip1 should be expected. In fact, we robustly observed co-transport
of RBP$^{cherry}$ and Aplip1$^{GFP}$ spots in both anterograde (*Figure 2C*, arrowhead; *Video 2*) and retrograde
(not shown) direction at a frequency consistent with the low frequency of single Aplip1$^{GFP}$ moving
particles (not shown). Furthermore, we observed BRP-short$^{straw}$ co-transport with Aplip1$^{GFP}$ (*Figure 2D*;

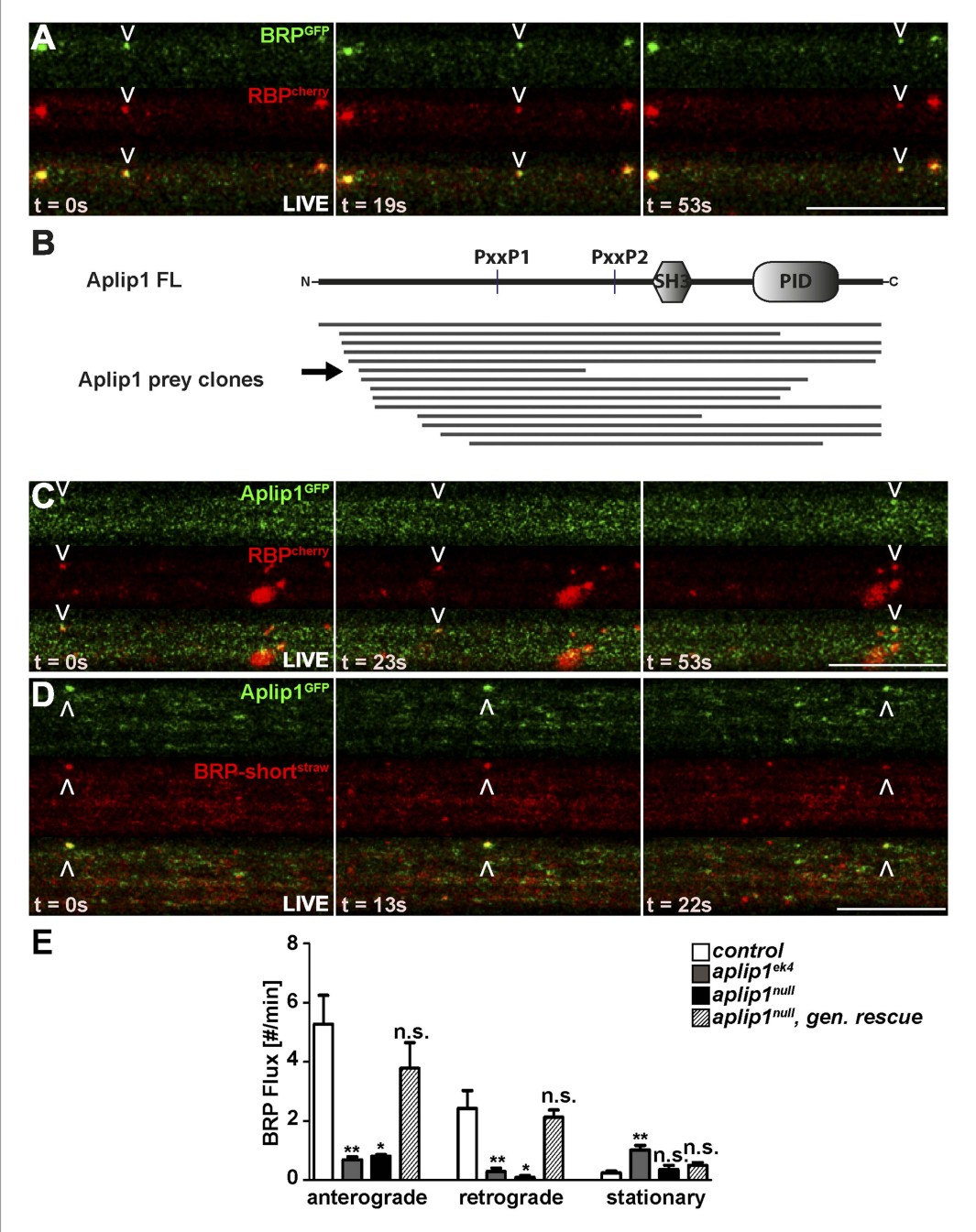

**Figure 2**. Live imaging of anterograde co-transport between BRP, RBP and APP-like protein interacting protein 1 (Aplip1). (**A**) Live imaging in intact third instar larvae showed anterograde co-transport of BRP$^{GFP}$ and RBP$^{cherry}$. See also, *Video 1*. (**B**) Schematic representation of Aplip1 domain structure containing two PxxP motifs, one Src-homology 3 (SH3) domain and one C-terminal phosphotyrosine interaction domain (PID) (FL = full-length). Lines represent Aplip1 prey fragments recovered in RBP SH3-II+III bait yeast-two-hybrid (Y2H) screen. Arrow indicates one single clone that contained only the first of the two Aplip1-PxxP motifs. (**C, D**) Live imaging in intact third instar larvae showed anterograde co-transport of Aplip1$^{GFP}$ and RBP$^{cherry}$ (**C**), as well as Aplip1$^{GFP}$ and BRP-short$^{straw}$ (**D**). Scale bars: (**A, C, D**) 10 µm. See also, *Videos 2, 3*. (**E**) Quantification of live imaging of BRP-short$^{straw}$ flux (spots passing through an axonal cross-section per minute) within the genetic backgrounds indicated. Anterograde and retrograde BRP-short$^{straw}$ flux was severely impaired in *aplip1$^{ek4}$* and *aplip1$^{null}$* mutant background, which was rescued when a genomic rescue construct for Aplip1 was introduced into the *aplip1$^{null}$* mutant background. BRP-short$^{straw}$ flux per min, control (n = 14 nerves): anterograde: 5.267 ± 0.975, retrograde: 2.423 ± 0.604, stationary: 0.241 ± 0.071; *aplip1$^{ek4}$* (n = 28 nerves): anterograde: 0.687 ± 0.098, retrograde: 0.284 ± 0.125, stationary: 1.023 ± 0.145; *aplip1$^{null}$*

*Figure 2. Continued*

(n = 11 nerves): anterograde: 0.808 ± 0.051, retrograde: 0.085 ± 0.064, stationary: 0.354 ± 0.148; *aplip1^{null}*, gen rescue (n = 26 nerves): anterograde: 3.783 ± 0.861, retrograde: 2.123 ± 0.239, stationary: 0.505 ± 0.084. All panels show mean values and errors bars representing SEM. *p ≤ 0.05; **p ≤ 0.01; ***p ≤ 0.001; ns, not significant, p > 0.05, Mann–Whitney U test.

---

*Video 3*), as expected with similarly low frequencies as observed for RBP/Aplip1 co-transport (not shown), further pointing towards a co-transport of RBP and BRP in conjunction with Aplip1. We used the live imaging assay to investigate BRP transport in different *aplip1* mutants to directly address whether removal of Aplip1 affects AZ scaffold protein transport. The *aplip1^{null}* allele completely and specifically removes the *aplip1* gene and was generated by P-element excision (*Klinedinst et al., 2013*). By comparison, the *aplip1^{ek4}* allele contains a point mutation in the C-terminal kinesin binding domain of Aplip1 that was shown to almost completely abolish the ability of Aplip1 to bind to KLC (*Horiuchi et al., 2005*). Anterograde and retrograde transport of BRP was drastically reduced compared to controls in both *aplip1* mutant alleles (*Figure 2E*). Through the introduction of a genomic (gen.) construct of Aplip1 into the *aplip1^{null}* mutant background (*aplip1^{null}*, gen. rescue), however, BRP flux (spots passing through an axonal cross-section in a given time) could be restored to WT level (*Figure 2E*). Quantification showed that retrograde transport in the *aplip1^{null}* mutant situation was somewhat more affected (27× less compared to control) than anterograde transport (7× less). Both directions appeared equally affected (about 8× less compared to controls) in the kinesin-binding defective *aplip1^{ek4}* mutant. It is noteworthy that the transport of SV cargo in the same mutant was reduced equally in both directions, whereas transport of mitochondria is only impaired in the retrograde direction (*Horiuchi et al., 2005*).

## RBP binds the transport adaptor Aplip1 via a high affinity PxxP-SH3 interaction

As our Y2H screen used the SH3-II and -III domains of RBP as bait (*Figure 3A*), PxxP motifs are expected to mediate the interaction with Aplip1. In fact, Aplip1 contains two PxxP motifs which were both present in most of the prey clones recovered in the Y2H screen, except for one single clone that contained only the first more N-terminal motif (*Figure 2B*, arrow). Using a semi-quantitative liquid Y2H assay and a set of Aplip1 constructs containing only either the first or the second PxxP motif (*Figure 3B*), we mapped the interaction between RBP and Aplip1 to the first of the two candidate PxxP motifs present in all clones isolated (*Figure 2B*). The second and third SH3 domain of RBP bound to this motif with comparable strength when measured with a semi-quantitative liquid Y2H assay (*Figure 3C*; mean ß-Gal4 units for: Aplip1-PxxP1/RBP SH3-II: 24.3 ± 6.6; Aplip1-PxxP1/RBPSH3-III: 29.1 ± 7.4; n = 3 independent experiments; mean ± SEM). No binding was observed between the second and third SH3 domains of RBP and Aplip1-PxxP2 (*Figure 3C*; mean ß-Gal4 units for: Aplip1-PxxP2/RBP SH3-II: 0.2 ± 0.0; Aplip1-PxxP2/RBPSH3-III: 0.2 ± 0.1; n = 3 independent experiments; mean ± SEM). When mutating either the PxxP1 motif of Aplip1 (P156 → A; P159 → A, giving rise to AxxA1) or introducing mutations known to interfere with PxxP ligand binding into the

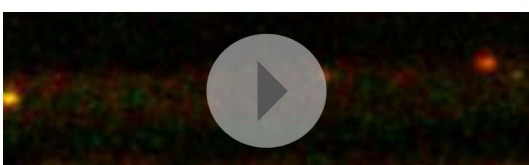

**Video 1.** Anterograde co-transport of BRP^{GFP} and RBP^{cherry}. Live imaging in intact third instar larvae showed anterograde co-transport of BRP^{GFP} and RBP^{cherry}. Video was captured at 0.6 s per frame and played back at 7× real time.

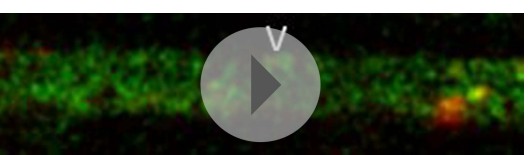

**Video 2.** Anterograde co-transport of Aplip1^{GFP} and RBP^{cherry}. Live imaging in intact third instar larvae showed anterograde co-transport of Aplip1^{GFP} and RBP^{cherry}. Video was captured at 0.6 s per frame and played back at 7× real time.

**Video 3.** Anterograde co-transport of Aplip1[GFP] and BRP-short[straw]. Live imaging in intact third instar larvae showed anterograde co-transport of Aplip1[GFP] and BRP-short[straw]. Video was captured at 0.414 s per frame and played back at 5x real time.

individual SH3 domains of RBP (SH3-II*/SH3-III*), the interaction was completely abolished (*Figure 3C*; mean ß-Gal4 units for: Aplip1-AxxA1/RBP SH3-II: 0.1 ± 0.1; Aplip1-AxxA1/RBP SH3-III: 0.2 ± 0.0; Aplip1-PxxP1/RBPSH3-II*: 0.1 ± 0.0; Aplip1-PxxP1/RBP SH3-III*: 0.1 ± 0.0; n = 3 independent experiments; mean ± SEM). We performed isothermal titration calorimetry (ITC) to measure the thermodynamics of the binding directly and compare Aplip1/RBP binding quantitatively to the established synaptic ligands of RBP. We used four different constructs, comprising either single RBP SH3 domains (I, II, and III) or a construct of two RBP SH3 domains (II+III) (see also *Figure 3A*). Whereas we could not detect any binding of the Aplip1 peptides to RBP SH3-I, we could determine $K_D$ constants for the single SH3-II, SH3-III and the tandem SH3-II+III (*Figure 3D*; *Figure 3—figure supplement 1*) domains of RBP. Both SH3-II and SH3-III single domains showed a binding affinity to Aplip1 peptides several fold stronger compared to either Cac, RIM1 or RIM2 (*Figure 3D*; *Figure 3—figure supplements 2–4*). However, the affinity of the Aplip1 peptides to the SH3-II+III domain was the highest observed which is indicative of co-operativity between both domains in peptide binding that could increase the local concentrations of Aplip1 at RBP binding pockets (BPs).

Finally, in order to get a deeper atomic insight into the structural basis of the binding of RBP towards Aplip1 in comparison to its synaptic ligands, we crystallized the *Drosophila* RBP SH3-II domain together with both an Aplip1 (*Figure 3E*; *Tables 1, 2, 3*) and a Cac peptide (*Figure 3F*; *Tables 1, 3, 4*), and RBP SH3-III with a Cac peptide (*Figure 3—figure supplement 5*; *Tables 1, 3*). *Drosophila* RBP SH3-II and -III share 49.2% sequence identity and adopt the canonical fold of SH3 domains (*Figure 3E,F*; *Figure 3—figure supplement 5*). Both domains superimpose with a root mean deviation of 0.8 Å for 64 pairs of Cα-atoms. Both peptides sequences harbor the canonical class I interaction motif +xΨPxxP (+, positively charged; x, any amino acid; Ψ hydrophobic amino acid, see *Figure 3D* for sequence) and are bound into the respective SH3 domain in 'plus' direction. We observed the classical poly-proline helix that allows for mainly hydrophobic protein-peptide interaction in all three structures. We detected the same hydrogen pattern between the protein side chains and peptide backbone in the structure of SH3-II with Aplip1 and Cac. The major difference is the side chain orientation of R1687 of Cac that π-stacks with its guanidinium function with Y1372, except for one copy, where it forms a salt-bridge to E1341. The equivalent residue to R1687 of Cac is R153 of the Aplip1 peptide, which forms, by contrast, a bidentate salt-bridge to D1336 (*Table 3*). A second major difference is induced by the two consecutive proline residues in the Cac peptide. Consequently, the peptide has a more polyproline type II conformation that brings T1692 closer to the protein surface and allows P1693 to deeper point in a hydrophobic pocket of the SH3-II domain. Whereas the C-terminal portion of the Aplip1 peptide is folded in a short $3_{10}$ helix, the N-terminus of the Aplip1 peptide adopts a random coil conformation with hydrophobic interactions to the surface of SH3-II. The Cac-derived peptide bound to SH3-III is fully defined in the electron density. However, the peptide main chain interaction with the SH3 domains is conserved. The side chain orientation of Cac R1687 is again different if bound to SH3-II or SH3-III. In complex with SH3-III, R1687 forms a bidentate hydrogen bond to SH3-III D1463 and E1648. A π-stacking interaction is not possible since Y1372 of SH3-II is replaced by SH3-III L1499. The central PxxP motifs of Aplip1 superimpose well in both structures if bound to SH3-II and SH3-III. Towards its C-terminus, the Aplip1-PxxP1 peptide adopts a slightly different random coil conformation compared to the structure when bound to SH3-II caused by two additional hydrogen bonds from T1692 and K1695 to the SH3-II domain (*Table 3*).

## The Aplip1-PxxP1 motif is needed for effective axonal RBP/BRP transport

Consistent with the idea that Aplip1 is mediating RBP transport, we found axonal aggregates consisting of both RBP and BRP in the *aplip1[ek4]*, as well as the *aplip1[null]* allele (*Figure 4B,C*). This ectopic

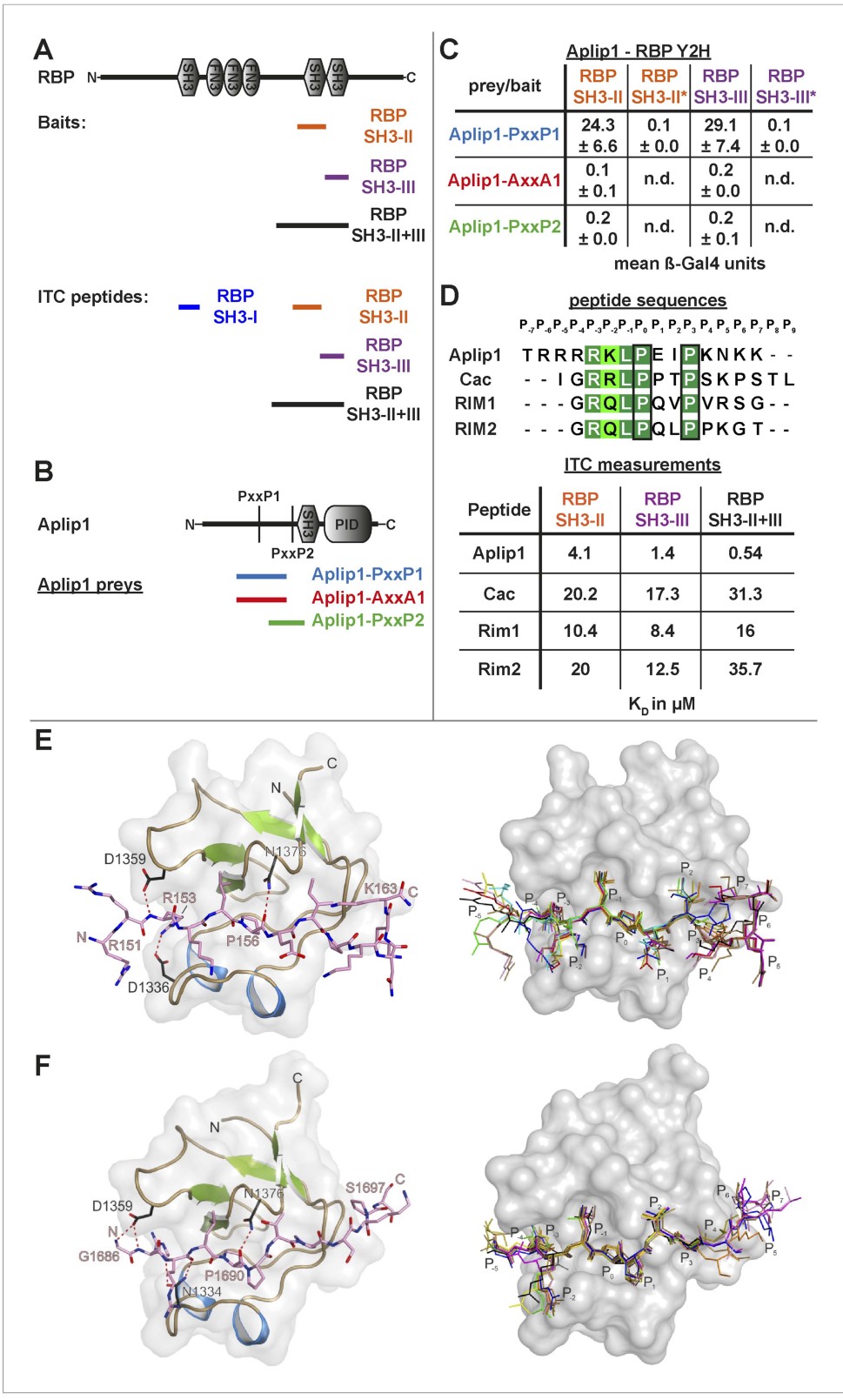

**Figure 3**. Aplip1 binds RBP using a high-affinity PxxP1-SH3 interaction. (**A**) Schematic representation of RBP domain structure containing three SH3 domains (I–III from the N-terminus) and three Fibronectin 3 (FN3) domains. The corresponding fragments used in the large scale Y2H screen (SH3-II+III) and used as bait (SH3-II and SH3-III) in the Y2H assay (**C**) against different Aplip1 prey constructs (**B**) are indicated. Different isothermal titration calorimetry
*Figure 3. continued on next page*

*Figure 3. Continued*

(ITC) peptides (SH3-I, SH3-II, SH3-III and SH3-II+III) used for ITC measurements (**D**) are also shown. (**B**) Schematic representation of Aplip1 domain structure entailing two PxxP motifs, one SH3 and one C-terminal PID. Different preys (Aplip1-PxxP1, -AxxA1 and -PxxP2) used in Y2H assay (**C**) are indicated. (**C**) Liquid Y2H assay of individual Aplip1 prey fragments against different RBP baits. Aplip1-PxxP1 interacted with both the single SH3-II and -III domains of RBP. Mutating this first PxxP motif (Aplip1-AxxA1) construct abolished the binding. Aplip1-PxxP2 showed no interaction to RBP SH3 domains. Constructs with point-mutated RBP SH3 domains (SH3-II*, SH3-III*) abolished the binding to Aplip1-PxxP1. (**D**) Peptide sequences used for ITC measurements. Aplip1 showed the strongest interaction with RBP compared with Cacophony (Cac), RIM1 and RIM2, with the strongest affinity (lowest $K_D$) between Aplip1 and the RBP SH3-II+III domain. (**E**, **F**) Crystal structure of Aplip1-peptide (**E**; see also, 3D for peptide sequence) and of Cac-peptide (**F**; see also, *Figure 3D* for peptide sequence) bound to RBP SH3-II. The SH3 domain is shown in gray surface representation with (left) and without (right) the respective protein in cartoon representation. The bound peptides are drawn in stick representation. Hydrogen bonds ≤3.3 Å are indicated by red dashes. In the right panel, several peptide SH3-II complexes as observed in the asymmetric unit are superimposed and shown in different colors. See also, *Tables 1–4*.

The following figure supplements are available for figure 3:

**Figure supplement 1**. ITC measurements for Aplip1 and RBP SH3 domains.

**Figure supplement 2**. ITC measurements for Cac and RBP SH3 domains.

**Figure supplement 3**. ITC measurements for RIM1 and RBP SH3 domains.

**Figure supplement 4**. ITC measurements for RIM2 and RBP SH3 domains.

**Figure supplement 5**. Crystal structure of Cac-peptide bound to RBP SH3-III domain.

---

RBP/BRP accumulation was rescued after introducing a genomic construct of Aplip1 into the *aplip1^null^* mutant background (*aplip1^null^*, gen. rescue; *Figure 4D*). Pan-neuronal expression of an Aplip1 cDNA equally rescued the axonal RBP/BRP accumulations (*Figure 4I*, quantification in K, L). Importantly, however, the expression of an Aplip-AxxA1 cDNA construct (integrated at the same chromosomal integration site as the control construct; expression and axonal presence confirmed with a newly generated Aplip1 Ab; not shown) could no longer rescue the RBP/BRP accumulation phenotype (*Figure 4J*, quantification in *Figure 4K,L*). Thus, we conclude that Aplip1 is involved in the transport of RBP/BRP to the AZ, whereby its functionality in this context largely depends on the integrity of its N-terminal PxxP1 motif.

## Aplip1 promotes BRP transport in absence of RBP

As indicated above, BRP accumulated in the axons of *aplip1* mutants as well. Thus, BRP could be transported through Aplip1 via binding to RBP, other yet undetected co-transported AZ proteins, or BRP could bind Aplip1 independently of RBP. We therefore created *aplip1/rbp* and *aplip1/brp* double mutants to investigate the functional relation of RBP and BRP with regard to Aplip1-dependent transport. While removing BRP in *srpk79D* mutants also abolished the axonal RBP spots (*Figure 5—figure supplement 1D*), removing BRP in *aplip1* mutants had no apparent effect on axonal RBP accumulations (*Figure 5B*; control in *Figure 5A*). On the other hand, genetic elimination of RBP did not interfere with the accumulation of BRP in *aplip1* mutant axons (*Figure 5E*; controls in *Figure 5C,D*). Thus, BRP transport also 'suffers' from the absence of the Aplip1 adaptor when RBP is removed in parallel. Hence, Aplip1 promotes BRP transport even in the absence of RBP. To address a putative molecular basis of this relationship, we performed a Y2H assay to test for direct interaction between five different BRP constructs and a full length Aplip1 construct (see *Figure 3B* for domain structure). Despite these efforts, robust interactions between Aplip1 and BRP fragments could not be detected (data not shown). Nonetheless, both RBP but also BRP were easily detected in anti-GFP immunoprecipitations (IPs) from a synaptic membrane preparation (*Figure 5F*; *Figure 5—figure supplement 2*) derived from *Drosophila*

**Table 1**. Data collection and refinement statistics

Data collection

| Structure | RBP SH3-II | RBP SH3-II | RBP SH3-III |
|---|---|---|---|
| | Aplip1 | Cac | Cac |
| PDB entry | 4Z88 | 4Z89 | 4Z8A |
| Space group | *C*2 | *P*2$_1$ | *I*222 |
| Wavelength (Å) | 0.91841 | 0.91841 | 0.91841 |
| Unit cell | | | |
| a; b; c (Å) | 108.3; 62.4; 163.6 | 58.3; 122.2; 68.5 | 52.1; 54.3; 73.6 |
| α; β; γ (°) | 90.0; 90.3; 90.0 | 90.0; 113.2; 90.0 | 90.0; 90.0; 90.0 |
| Resolution (Å)* | 50.00–2.09 | 50.00–2.64 | 50.00–1.75 |
| | (2.19–2.09) | (2.74–2.64) | (1.86–1.75) |
| Unique reflections | 64,269 (7760) | 25,229 (2591) | 10,690 (1579) |
| Completeness* | 98.9 (92.4) | 96.9 (95.0) | 98.7 (92.6) |
| $<I/\sigma(I)>$* | 7.7 (2.6) | 8.0 (2.1) | 14.2 (2.2) |
| $R_{meas}$*, † | 0.127 (0.533) | 0.157 (0.726) | 0.127 (0.663) |
| $CC_{1/2}$* | 99.1 (68.0) | 98.9 (81.2) | 99.7 (76.5) |
| Redundancy* | 3.7 (3.7) | 3.5 (3.2) | 5.6 (3.1) |
| Refinement | | | |
| Non-hydrogen atoms | 7564 | 6239 | 850 |
| $R_{work}$*, ‡ | 0.210 (0.314) | 0.255 (0.367) | 0.159 (0.233) |
| $R_{free}$*, § | 0.236 (0.396) | 0.312 (0.490) | 0.208 (0.332) |
| Average B-factor (Å$^2$) | 40.8 | 52.10 | 18.8 |
| No. of complexes | 24 | 10 | 1 |
| Protein residues | 6484/41.0 | 663/51.1 | 74/17.6 |
| Peptide residues | 861/42.7 | 92/63.6 | 15/15.9 |
| Buffer molecules | 11/40.2 | 1/46.3 | – |
| Water molecules | 57/29.6 | 134/30.3 | 110/28.6 |
| r.m.s.d.# | | | |
| bond length (Å) | 0.007 | 0.005 | 0.010 |
| bond angles (°) | 1.224 | 1.140 | 1.210 |
| Ramachandran outliers (%) | 0.1 | 0.56 | 0 |
| Ramachandran favoured (%) | 98.4 | 98.0 | 100 |

*values in parentheses refer to the highest resolution shell.

†$R_{meas} = \Sigma_h [n/(n-1)]^{1/2} \Sigma_i |I_h - I_{h,i}|/\Sigma_h \Sigma_i I_{h,i}$ where $I_h$ is the mean intensity of symmetry-equivalent reflections and $n$ is the redundancy.

‡$R_{work} = \Sigma_h |F_o - F_c|/\Sigma F_o$ (working set, no σ cut-off applied).

§$R_{free}$ is the same as $R_{work}$, but calculated on 5% of the data excluded from refinement.

#Root-mean-square deviation (r.m.s.d.) from target geometries.

CC, coiled coil.

head extracts of pan-neuronal driven Aplip1-GFP cDNA construct (*Depner et al., 2014*). Of note, within axons of *rbp^null* mutant larvae, ectopic BRP accumulations could not be observed (not shown). Thus, we provide evidence for an RBP-independent but Aplip1-dependent transport component for BRP, whose mechanistic details have still to be deciphered. Taken together, our results imply that though BRP and RBP are co-transported in the WT situation, their Aplip1-dependent transport can be genetically uncoupled.

**Table 2.** Completeness of the model for RBP SH3-II and bound Aplip1 peptide

| RBP SH3-II | Range | Aplip1 | Range |
|---|---|---|---|
| chain A | 1318–1382 | chain M | 153–163 |
| chain B | ×1318–1382 | chain N | 155–159 |
| chain C | ×1318–1381 | chain O | 154–163 |
| chain D | ×1318–1382 | chain P | 153–159 |
| chain E | 1319–1381 | chain Q | 151–163 |
| chain F | ×1318–1380 | chain R | 153–159 |
| chain G | ×1318–1381 | chain S | 151–163 |
| chain H | ×1318–1382 | chain T | 152–156 |
| chain I | ×1318–1382 | chain U | 152–163 |
| chain J | ×1318–1381 | chain V | 152–158 |
| chain K | ×1318–1381 | chain W | 152–163 |
| chain L | ×1318–1381 | chain X | 152–158 |

Completeness of the model given for the 12 complexes of RBP SH3-II bound to the Aplip1 peptide [149]TRRRRKLPEIPKNKK[163]. Superscript 'x' indicates additional N-terminal residues of RBP SH3-II originating from the linker region between the protease cleavage site and the N-terminus.

## RBP and BRP form ectopic AZs at the axonal plasma membrane of *aplip1* mutants

The BRP flux in axons of *aplip1* mutants was severely diminished, but not completely abolished (*Figure 2E*). At the same time, AZ localization of both BRP and RBP at synaptic terminals of *aplip1* mutants was still observed in both *aplip1* alleles (not shown), although slightly reduced (not shown). This indicates that alternative transport mechanisms and adaptors exist which operate in parallel to Aplip1, as the synaptic phenotype is relatively weak. In fact, axonal accumulations of BRP have already been described for Acyl-CoA long-chain Synthetase (Acsl, *Liu et al., 2011b*) as well as for Unc-51 (Atg1) mutants (*Wairkar et al., 2009*). In our experiments, we found RBP to invariably co-cluster with BRP in the mutants mentioned (*Figure 6B,C*; control in *Figure 6A*), and equally in mutants of the *Drosophila* ß-amyloid protein precursor-like (Appl; *Torroja et al., 1999a, 1999b*; *Figure 6D*) and Unc-76 (*Gindhart et al., 2003*; *Figure 6E*). The fact that RBP and BRP tightly co-accumulated in axonal aggregates of all these transport mutants strengthens the probability that BRP is always co-transported with RBP.

To gain a deeper insight into the substructure of the BRP/RBP accumulations in *aplip1* mutant axons, we again used two-colour STED microscopy. In contrast to the srpk79D aggregates, however, STED images of axonal BRB/RBP accumulations were reminiscent of mature synaptic AZs (*Liu et al. 2011a*), with BRP[C-term] signal surrounding the RBP signal, which, in turn, is oriented closer towards the axonal plasma membrane (*Figure 7A*, arrow head; plasma membrane indicated by dashed line). Interestingly, in contrast to the floating T-bar super-aggregates in srpk79D mutants (*Johnson et al., 2009*; *Nieratschker et al., 2009*), these axonal BRP spots in *aplip1* mutants were positive for Syd-1 (compare *Figures 1C, 7B*). Intriguingly, floating T-bars have been observed in synaptic boutons in *syd-1* mutants (*Owald et al., 2010*). Together, this is suggestive of a role of Syd-1 in the membrane-anchoring of AZ proteins.

Furthermore, we asked whether BRP/RBP aggregates identified in *aplip1* mutants represent ectopic AZs forming at the axonal plasma membrane. In fact, EM analysis easily revealed T-bar structures, typical for synaptic terminals (*Figure 7C*, arrow heads, magnification in E), at axonal plasma membranes of *aplip1* mutants (*Figure 7D*, arrow heads, magnification in F), but never in controls (not shown). We found these ectopic axonal T-bars surrounded by SV profiles (*Figure 7D*, arrows), very similar to 'normally positioned' T-bars at the presynaptic terminal (*Figure 7C*, arrows). Consistently, the SV marker Synaptotagmin-1 (Syt-1) was found to be associated with BRP/RBP accumulations in *aplip1[null]* mutants (*Figure 7H*, quantification in *Figure 7K*). This phenotype could be rescued by the expression of an Aplip1 WT cDNA construct (*Figure 7I*, quantification in *Figure 7K*) but not by the expression of the Aplip1-AxxA1 construct (*Figure 7J*; quantification in *Figure 7K*). Thus, a point-like interaction surface of Aplip1 which binds RBP with high affinity is important to block a whole sequence of assembly events at the axonal plasma membrane, including AZ scaffold ('T-bar') formation and the accumulation of SVs.

To further support the importance of adaptor protein—cargo interaction in blocking ectopic AZ assembly we downregulated the expression of motor proteins. This also leads to transport defects and ectopic axonal AZ protein accumulations but in principle leaving the adaptor protein—cargo interaction intact. Interestingly, motoneuronal driven Imac-RNAi led to only few axonal BRP/RBP accumulations although with no preference concerning their direction in relation to the axonal plasma membrane (*Figure 7—figure supplement 1B*; arrow heads).

**Table 3.** Hydrogen bonding interaction

| Aplip1 | SH3-II | Distance |
|---|---|---|
| Arg153[N] | Asp1359[OD2] | 2.4 |
| Arg153[NH2] | Asp1336[OD1] | 3.0 |
| Arg153[NH2] | Asp1336[OD2] | 2.6 |
| Lys154[N] | Asn1334[OD1] | 2.9 |
| Lys154[O] | Asn1334[ND2] | 3.0 |
| Pro156[O] | Asn1376[ND2] | 2.8 |
| **Cac** | **SH3-II** | **Distance** |
| Gly1686[N] | Asp1359[OD2] | 2.7 |
| Arg1687[N] | Asp1359[OD2] | 2.8 |
| Arg1688[N] | Asn1334[OD1] | 3.0 |
| Arg1688[O] | Asn1334[ND2] | 2.9 |
| Pro1690[O] | Asn1376[ND2] | 2.8 |
| **Cac** | **SH3-III** | **Distance** |
| Arg1687[NH1] | Asp1463[OD1] | 2.9 |
| Arg1687[NH1] | Glu1488[OE2] | 3.0 |
| Arg1687[NH2] | Glu1488[OE2] | 3.1 |
| Arg1688[N] | Asn1461[OD1] | 2.8 |
| Arg1688[O] | Asn1461[ND2] | 3.0 |
| Pro1690[O] | Asn1376[ND2] | 2.9 |
| Thr1692[OG] | Asn1376[ND2] | 2.9 |
| Lys1695[O] | Tyr1451[OH] | 2.8 |
| Ser1697[OG] | Leu1450[O] | 2.7 |

Hydrogen bonding interaction of RBP SH3-II with Aplip1 and Cac, as well as RBP SH3-III in complex with Cac. Distance ≤3.2 Å are given in Å.

In contrast motoneuronal driven KHC-RNAi showed prominent axonal aggregates consistent of BRP/RBP but most of the time showing an irregular, elongated shape (*Figure 7—figure supplement 1C*; arrow heads). As mentioned above, proper T-bars were identified in *aplip1* mutant axons with ease. In contrast, systematic EM analysis of *khc* mutant axons revealed just one electron dense material that showed a T-bar-like appearance (*Figure 7—figure supplement 1D*; arrow head, magnifications in E, F) but never in control (ctrl) or motoneuronal driven Imac-RNAi.

In summary, we find that the SH3-II and -III interaction surface of RBP serves as a multi-functional platform for differential protein interaction with either other AZ components or the transport adaptor and therefore, motor-cargo linkage. Thus, interaction surfaces of RBP/BRP 'cargo complexes' might be shielded and blocked from undergoing premature assembly by interactions with transport adaptors, while genetically induced loss of these adaptors might provoke premature AZ assembly.

## Discussion

Large multi-domain scaffold proteins such as BRP/RBP are ultimately destined to form stable scaffolds, characterized by remarkable tenacity and a low turnover, likely due to stabilization by multiple homo- and heterotypic interactions simultaneously (*Sigrist and Schmitz, 2011*). How these large and 'sticky' AZ scaffold components engage into axonal transport processes to ensure their 'safe' arrival at the synaptic terminal remains to be addressed. We find here that the AZ scaffold protein RBP binds the transport adaptor Aplip1 using a 'classic' PxxP/SH3 interaction. Notably, the same RBP SH3 domain (II and III) interaction surfaces are used for binding the synaptic AZ ligands of RBP, that is, RIM and the voltage gated Ca$^{2+}$ channel (*Wang et al., 2002*; *Kaeser et al., 2011*; *Liu et al., 2011a*; *Davydova et al., 2014*), though with clearly lower affinity than for Aplip1. A point mutation which disrupts the Aplip1-RBP interaction provoked a 'premature' capture of RBP and the co-transported BRP at the axonal membrane, thus forming ectopic but, concerning T-bar shape and BRP/RBP arrangement, WT-like AZ scaffolds. The Aplip1 orthologue Jip1 has been shown to homo-dimerize via interaction of its SH3 domain (*Kristensen et al., 2006*). Thus, the multiplicity of interactions, with Aplip1 dimers binding to two SH3 domains of RBP as well as to KLC, might form transport complexes of sufficient avidity to ensure tight adaptor–cargo interaction and prevent premature capture of the scaffold components.

Our intravital imaging experiments showed that within axons RBP and BRP are co-transport in shared complexes together with Aplip1, whereas we, despite efforts, were unable to detect any co-transport of other AZ scaffold components, that is, Syd-1 or Liprin-α with BRP/RBP (not shown). In addition, STED analysis of axonal aggregates in *srpk79D* mutants showed BRP/RBP in stoichiometric amounts, but also failed to detect other AZ scaffold components. Moreover, BRP and RBP co-aggregated in the axoplasm of several other transport mutants we tested (*acsl, unc-51, appl, unc-76*), consistent with both proteins entering synaptic AZ assembly from a common transport complex. Of note, during AZ assembly at the NMJ, BRP incorporation is invariably delayed compared to the 'early assembly' phase which is driven by the accumulation of Syd-1/Liprin-α scaffolds (*Fouquet et al., 2009*;

**Table 4.** Completeness of the model for RBP SH3-II and bound Cac peptide

| RBP SH3-II | Range | Cac | Range |
|---|---|---|---|
| chain A | 1318–1381 | chain a | 1686–1697 |
| chain B | ×1318–1381 | chain b | 1686–1695 |
| chain C | ×1318–1382 | chain c | 1686–1697 |
| chain D | ×1318–1381 | chain d | 1686–1697 |
| chain E | 1318–1382 | chain e | 1685–1694 |
| chain F | ×1318–1382 | chain f | 1685–1693 |
| chain G | ×1318–1382 | chain g | 1686–1693 |
| chain H | 1318–1381 | chain h | 1686–1693 |
| chain I | ×1318–1381 | chain i | 1686–1693 |
| chain J | ×1318–1382 | chain j | 1686–1697 |

Completeness of the model given for the six complexes of RBP SH3-II and the bound Cac peptide [1685]IGRRLPPTPSKPSTL[1699]. Superscript 'x' indicates additional N-terminal residues of RBP SH3-II originating from the linker region between the protease cleavage site and the N-terminus.

Owald et al., 2010, 2012). As the early assembly phase is, per se, still reversible (Owald et al., 2010), the transport of 'stoichiometric RBP/BRP complexes' delivering building blocks for the 'mature scaffold' might drive AZ assembly into a mature, irreversible state (Owald et al., 2010), and seems mechanistically distinct from early scaffold assembly mechanisms.

Previous work suggested that AZ scaffold components (Piccolo, Bassoon, Munc-13 and ELKS) in rodent neurons are transported to assembling synapses as 'preformed complexes', so-called Piccolo-Bassoon-Transport Vesicles (PTVs; Zhai et al., 2001; Shapira et al., 2003; Maas et al., 2012). The PTVs are thought to be co-transported with SV precursors (Ahmari et al., 2000; Tao-Cheng, 2007; Bury and Sabo, 2011) anterogradely mediated via a KHC(KIF5B)/ Syntabuli/Syntaxin-1 complex (Cai et al., 2007) and retrogradely via a direct interaction between Dynein light chain and Bassoon (Fejtova et al., 2009). Since their initial description, however, further investigations of PTVs have been hampered by the apparent relative scarcity of PTVs, and by the lack of genetic or biochemical options for specifically interfering with their transport or final incorporation into AZs.

Despite efforts we were not able to detect a direct interaction of Aplip1 and BRP although their common transport can be uncoupled from the presence of RBP. One possible explanation could be a direct interaction of Aplip1 to other AZ proteins that are co-transported together with BRP and RBP. It is interesting that the very C-terminus of BRP is essential for SV clustering around the BRP-based AZ cytomatrix (Hallerman et al., 2010). Thus, it is tempting to speculate that adaptor/transport complex binding might block premature AZ protein/SV interactions before AZ assembly, but further analysis will have to await more atomic details as we could gain for the RBP::Aplip1 interaction.

The down-regulation of the motor protein KHC also provoked severe axonal co-accumulations of BRP and RBP but per se should leave the adaptor protein-AZ cargo interaction intact. In contrast to aplip1, the axonal aggregations in khc mutants adapted irregular shapes most of the time, likely not representing T-bar-like structures. Thus, our data suggest a mechanistic difference when comparing the consequences between eliminating adaptor cargo interactions with a direct impairment of motor functions. Still, we cannot exclude that trafficking of AZ complexes naturally antagonizes their ability to assemble into T-bars.

The idea that proteins/molecules are held in an inactive state till they reach their final target has been observed in many other cell types. For example, in the context of local translation control, mRNAs are shielded or hidden in messenger ribonucleoprotein particles during transport so that they are withheld from cellular processing events such as translation and degradation. Shielding is thought to operate through proteins that bind to the mRNA and alter its conformation while at the correct time or place the masking protein is influenced by a signal that alleviates its shielding effect (Spirin, 1996; Johnstone and Lasko, 2001). As another example, hydrolytic enzymes, for example, lysosomes, are transported as proteolytically inactive precursors that become matured by proteolytic processing only within late endosomes or lysosomes (Ishidoh and Kominami, 2002). Particularly relevant in the context of AZ proteins involved in exocytosis, the $H_{abc}$ domain of Syntaxin-1 folds back on the central helix of the SNARE motif to generate a closed and inactive conformation which might prevent the interaction of Syntaxin-1 with other AZ proteins during diffusion (Dulubova et al., 1999; Ribrault et al., 2011).

Previously, genetic analysis of C. elegans axons forming en passant synapses suggested a tight balance between capture and dissociation of protein transport complexes to ensure proper positioning of presynaptic AZs. In this study, overexpression of the kinesin motor Unc-104/KIF1A

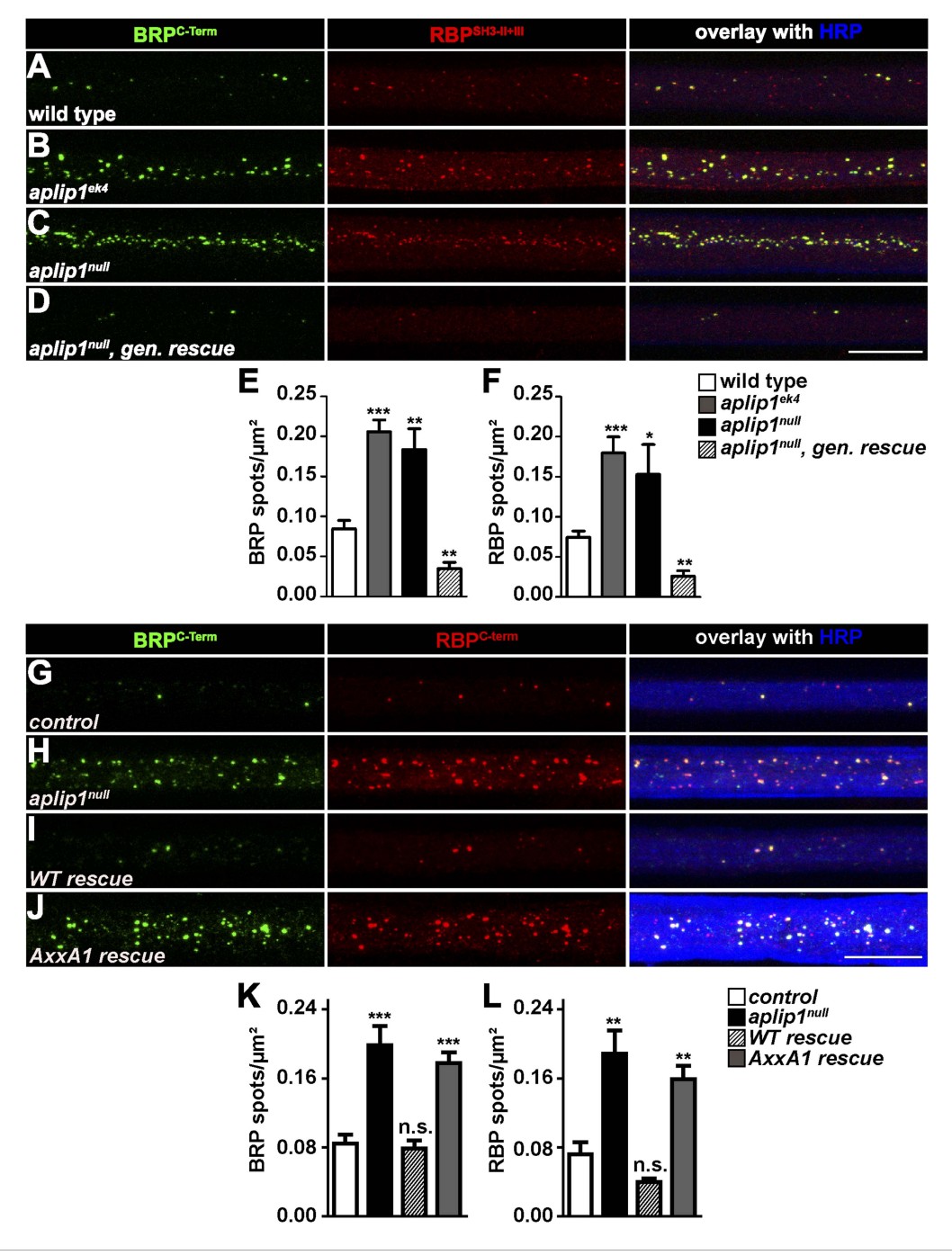

**Figure 4**. Aplip1-PXXP1 motif is needed for its function as RBP/BRP transport adaptor. (**A–D**) Nerve bundles of segments A1–A3 from third instar larvae of the genotypes indicated labeled with the Abs indicated. (**E, F**) Quantification of BRP/RBP spot numbers. BRP spots per $\mu m^2$: WT (n = 8 nerves): 0.084 ± 0.010; *aplip1*[ek4] (n = 9 nerves): 0.205 ± 0.025; *aplip1*[null] (n = 8 nerves): 0.183 ± 0.025; *aplip1*[null], *gen. rescue* (n = 8 nerves): 0.034 ± 0.007; RBP spots per $\mu m^2$, WT (n = 8 nerves): 0.074 ± 0.007; *aplip1*[ek4] (n = 9 nerves): 0.180 ± 0.019; *aplip1*[null] (n = 8 nerves): 0.153 ± 0.037; *aplip1*[null], *gen. rescue* (n = 8 nerves): 0.025 ± 0.006. All panels show mean values and errors bars representing SEM. *p ≤ 0.05; **p ≤ 0.01; ***p ≤ 0.001; ns, not significant, p > 0.05, Mann–Whitney U test. (**G–J**) Nerve bundles of segment A1–A3 from third instar larvae of the genotypes indicated labeled with the Abs indicated. BRP and RBP co-localized in control animals and accumulated in a co-localizing fashion in axons of *aplip1*[null] mutant animals. Re-expression of an Aplip1-WT cDNA construct in the *aplip1*[null] background rescued the phenotype, while re-expression of an AxxA1 construct did not.
*Figure 4. continued on next page*

*Figure 4. Continued*

(**K**, **L**) Quantification of the number of BRP/RBP spots per $\mu m^2$ axon. BRP spots per $\mu m^2$, control (n = 12 nerves): 0.084 ± 0.010; *aplip1$^{null}$* (n = 16 nerves): 0.198 ± 0.022; *WT rescue* (n = 14 nerves): 0.078 ± 0.009; *AxxA1 rescue* (n = 14 nerves): 0.177 ± 0.012; RBP spots per $\mu m^2$, control (n = 12 nerves): 0.071 ± 0.013; *aplip1$^{null}$* (n = 16 nerves): 0.188 ± 0.026; *WT rescue* (n = 14 nerves): 0.039 ± 0.004; *AxxA1 rescue* (n = 14 nerves): 0.158 ± 0.015. All panels show mean values and errors bars representing SEM. *p ≤ 0.05; **p ≤ 0.01; ***p ≤ 0.001; ns, not significant, p > 0.05, Mann–Whitney U test. Scale bar: (**A–D**, **G–J**) 10 μm.

reduced the capture rate and could suppress the premature axonal accumulations of AZ/SV proteins in mutants of the small, ARF-family G-protein Arl-8. Interestingly, large axonal accumulations in *arl-8* mutants displayed a particularly high capture rate (*Klassen et al., 2010*; *Wu et al., 2013*). Similarly, both *aplip1* alleles exhibited enlarged axonal BRP/RBP accumulations. Thus, the capture/dissociation balance for AZ components might be shifted towards 'capture' in these mutants, consistent with the ectopic axonal T-bar formation. It is tempting to speculate that loss of Aplip1-dependent scaffolding and/or kinesin binding provokes the exposure of critical 'sticky' patches of scaffold components such as RBP and BRP. Such opening of interaction surfaces might increase 'premature' interactions of cargo proteins actually destined for AZ assembly, thus increase overall size of the cargo complexes by oligomerization between AZ proteins and, finally, promote premature capture and ultimately ectopic AZ-like assembly. On the other hand, the need for the system to unload the AZ cargo at places of physiological assembly (i.e., presynaptic AZ) might pose a limit to the 'wrapping' of AZ components and ask for a fine-tuned capture/dissociation balance.

Several mechanisms for motor/cargo separation such as (i) conformational changes induced by guanosine-5′-triphosphate hydrolysis, (ii) posttranslational modification as de/phosphorylation, or (iii) acetylation affecting motor-tubulin affinity, have been suggested for cargo unloading (*Hirokawa et al., 2010*). Notably, Aplip1 also functions as a scaffold for JNK pathway kinases, whose activity causes motor-cargo dissociation. JNK probably converges with a mitogen-activated protein kinase (MAPK) cascade (MAPK kinase kinase Wallenda phosphorylating MAPK kinase Hemipterous) in the phosphorylation of Aplip1, thereby dissociating Aplip1 from KLC. Thus, JNK signaling, co-ordinated by the Aplip1 scaffold, provides an attractive candidate mechanism for local unloading of SVs (*Horiuchi et al., 2007*) and, as shown here, AZ cargo at synaptic boutons. Our study further emphasises the role of the Aplip1 adaptor, whose direct scaffolding role through binding AZ proteins might well be integrated with upstream controls via JNK and MAP kinases. Intravital imaging in combination with genetics of newly assembling NMJ synapses should be ideally suited to further dissect the obviously delicate interplay between local cues mediating capturing and axonal transport with motor-cargo dissociation.

## Materials and methods

### Genetics

Fly strains were reared under standard laboratory conditions (*Sigrist et al., 2003*) on semi-defined medium (Bloomington recipe). For all experiments both male and female larvae were used for analysis. The following genotypes were used: WT: +/+ (*w1118*). srpk79D: *srpk79D$^{atc}$/srpk79D$^{atc}$* (unless otherwise noted). *srpk79D$^{vn}$*: *srpk79D$^{vn}$/srpk79D$^{vn}$*. *srpk79D$^{atc}$*: *srpk79D$^{atc}$/srpk79D$^{atc}$*. *brp$^{Df}$/+; srpk79D*: *Df(2R)BSC29/+; srpk79D$^{atc}$/srpk79D$^{atc}$*. *brp$^{null}$/brp$^{Df}$; srpk79D*: *brp$^{69}$/Df(2R)BSC29; srpk79D$^{atc}$/srpk79D$^{atc}$*. *rbp$^{Df}$/+;srpk79D*: *Df(3R)S2.01/+; srpk79D$^{atc}$/srpk79D$^{atc}$*. *rbp$^{null}$/rbp$^{Df}$; srpk79D*: *rbp$^{STOP1}$/Df(3R)S201; srpk79D$^{atc}$/srpk79D$^{atc}$*. *aplip1$^{ek4}$*: *aplip1$^{ek4}$/aplip1$^{ek4}$*. *aplip1$^{null}$*: *aplip1$^{ex213}$/aplip1$^{ex213}$*. *aplip1, gen.rescue*: *aplip1$^{gen.rescue(ex213)}$/aplip1$^{gen.rescue(ex213)}$*. Aplip1 cDNA rescue: control: *elav/+;; aplip1$^{ex213}$/+*. *aplip1$^{null}$*: *elav/+;;aplip1$^{ex213}$/aplip1$^{ex213}$*. WT rescue: *elav/+;UAS-Aplip1-WT/+;aplip1$^{ex213}$/aplip1$^{ex213}$*. AxxA1 rescue: *elav/+;UAS-Aplip1-AxxA1/+;aplip1$^{ex213}$/aplip1$^{ex213}$*. *brp$^{Df}$/+;aplip1$^{ek4}$*: *Df(2R)BSC29/+; aplip1$^{ek4}$/aplip1$^{ek4}$*. *brp$^{null}$/brp$^{Df}$;aplip1$^{ek4}$*: *brp$^{69}$/Df(2R)BSC29; aplip1$^{ek4}$/aplip1$^{ek4}$*. Ok6>+: OK6-Gal4/+. OK6>Aplip1-RNAi;rbp$^{Df}$/+: OK6-Gal4/UAS-*aplip1*-RNAi;Df(3R)S2.01/+. OK6>Aplip1-RNAi; *rbp$^{null}$/$^{Df}$*: OK6-Gal4/UAS-*aplip1*-RNAi; *rbp$^{STOP1}$/Df(3R)S201*. acsl: *acsl$^{05847}$/acsl$^1$*. unc51 (atg-1): *atg1$^{ey07351}$/ Df(3L)BSC10*. appl: *appl$^{BG0264}$/appl$^{Df(1)yT7-518}$*. unc-76: *unc-76$^{G0158}$/y*. Aplip1$^{GFP}$,BRP-short$^{straw}$: OK6-Gal4/UAS-BRP-short$^{straw}$;UAS-Aplip1$^{GFP}$/+. Aplip1$^{GFP}$,RBP$^{cherry}$: OK6-Gal4/OK6-Gal4;UAS-Aplip1$^{GFP}$/UAS-Aplip1$^{GFP}$

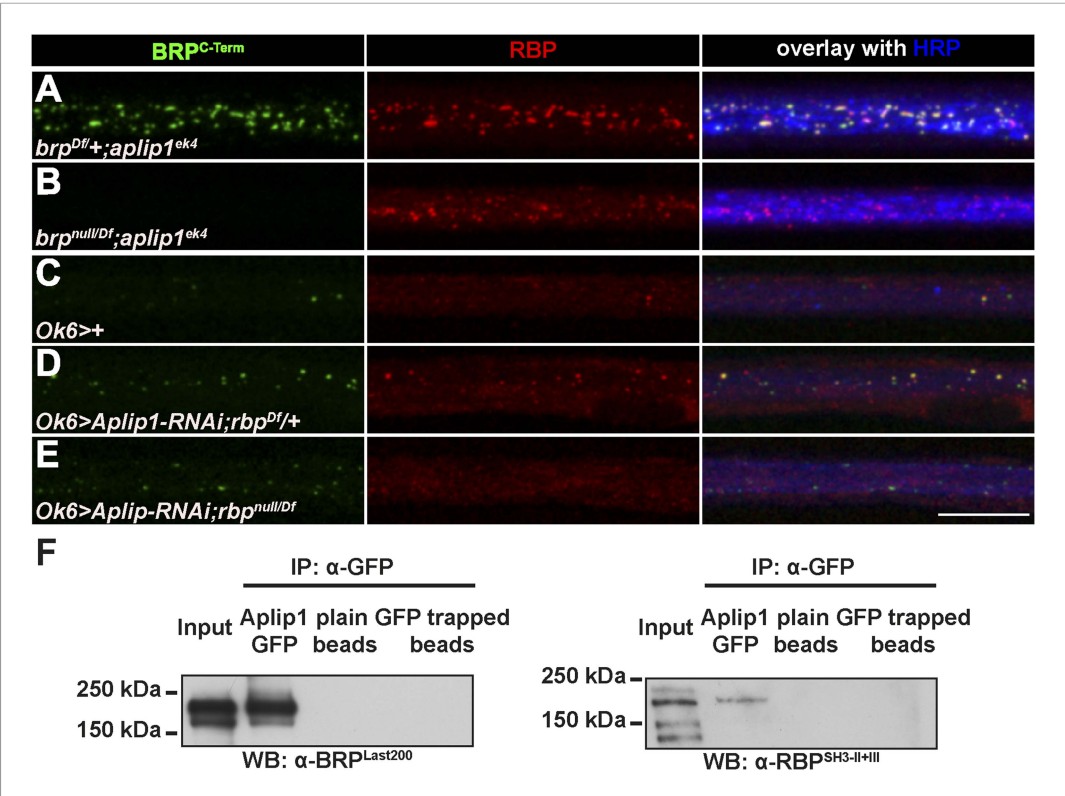

**Figure 5**. Aplip1 promotes BRP transport in absence of RBP. (**A–E**) Nerve bundles of segments A1–A3 from third instar larvae of the genotypes indicated labeled with the Abs indicated. (**A**) Removing one copy of BRP in *aplip1^ek4* mutants had no apparent effect on axonal RBP accumulation. (**B**) RBP still accumulates in *brp^null*;*aplip1^ek4* double mutants. (**C, D**) Driver control and removing one copy of RBP in motoneuronal driven Aplip1-RNAi had no apparent effect on axonal BRP accumulation. (**E**) BRP still accumulates in rbp^null^,*aplip1* double mutants Scale bar: (**A–E**) 10 μm. (**F**) Immunoprecipitation (IP) of Aplip1^GFP with anti-GFP Ab from Drosophila active zone (AZ) protein-enriched fraction was followed by Western blot (WB) analysis using anti-BRP^Last200 and anti-RBP^SH3-II+III. Both BRP and RBP could be detected in Aplip1^GFP IPs, but are absent in controls (plain beads; GFP trapped beads). (For whole WBs, see *Figure 5—figure supplement 2*).

The following figure supplements are available for figure 5:

**Figure supplement 1**. Accumulation of BRP in *srpk79D* mutant axons is unaffected by removing RBP.

**Figure supplement 2**. IP of Aplip1^GFP with anti-GFP (Full blot).

were crossed to upstream activator sequence (UAS)-RBP^cherry/UAS-RBP^cherry. BRP^GFP,RBP^cherry: OK6-Gal4/ OK6-Gal4;genomicBRP^GFP/genomicBRP^GFP were crossed to UAS-RBP^cherry/UAS-RBP^cherry. Live imaging BRP-short^straw in *aplip1* mutant backgrounds (*Figure 2E*): ctrl: OK6-Gal4/UAS-BRP-short^straw.*aplip1^ek4*: OK6-Gal4/UAS-BRP-short^straw;*aplip1^ek4*/*aplip1^ek4*. *aplip1^null*: OK6-Gal4/UAS-BRP-short^straw;*aplip1^ex213*/*aplip1^ex213*. *aplip1^gen.rescue*: OK6-Gal4/UAS-BRP-short^straw;*aplip1^gen.rescue(ex213)*/ *aplip1^gen.rescue(ex213)*. Ok6/+;UAS-KHC-RNAi. Ok6/+;UAS-Imac-RNAi.

Stocks were obtained from: brp^69 (**Kittel et al., 2006**), Df(3R)S2.01 and rbp^STOP1 (**Liu et al., 2011a**), *aplip1^ex213* and *aplip1^gen.rescue(ex213)* gift from Catherine Collins (**Klinedinst et al., 2013**), srpk79D^atc (**Johnson et al., 2009**), srpk79D^vn (**Nieratschker et al., 2009**), UAS-Aplip1^GFP (**Horiuchi et al., 2005**), UAS-BRP-short^straw (**Schmid et al., 2008**) and genomic BRP^GFP (**Matkovic et al., 2013**). The *aplip1^ek4*, Df(2R)BSC29, acsl^05847, acsl^1, atg1^ey07351, appl^BG0264, appl ^Df(1)yT7-518, Df(3L)BSC10, unc-76^G0158 lines were provided by the Bloomington Drosophila Stock Center. UAS-Aplip1-RNAi, UAS-Imac-RNAi and UAS-KHC-RNAi from VDRC.

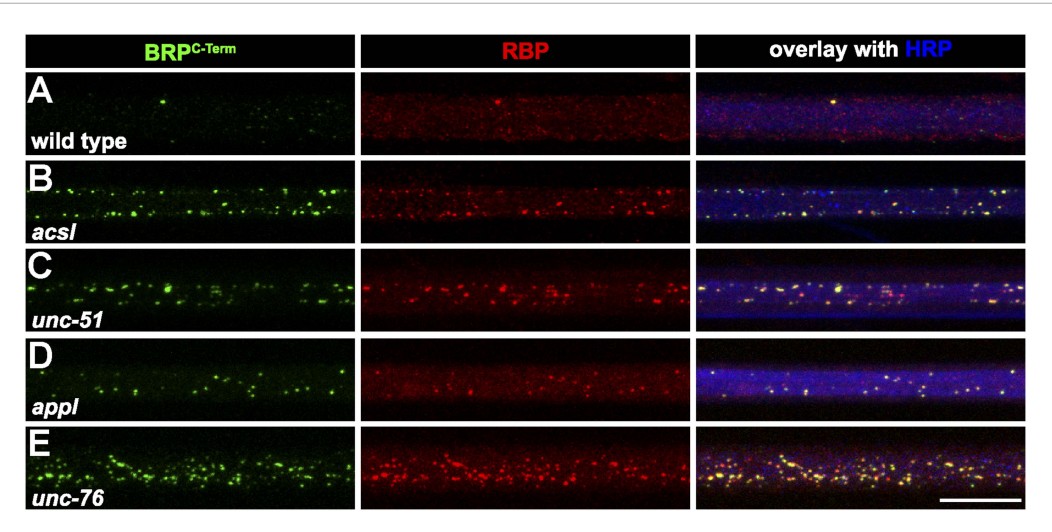

**Figure 6**. Several known transport adaptor mutants showed axonal BRP and RBP co-accumulations. (**A–E**) Nerve bundles of segment A1–A3 from third instar larvae of the genotypes indicated labeled with the Abs indicated. BRP and RBP accumulated in a co-localizing manner in axons of WT (**A**), acsl (**B**), unc-51 (atg-1; **C**), appl (**D**) and unc-76 (**E**). Scale bar: 10 μm.

## Generation of RBP^cherry^ cDNA construct

RBP cDNA was assembled based on exon annotation sequence of RBP-PF isoform from flybase. cDNA clones, AT04807; RH38268 and a gene synthesis fragment from MWG eurofins GMBH, Germany, containing 1–1131 bp of RBP-PF isoform were used to assemble the cDNA. All the fragments were cloned into a modified pENTR4 cloning vector described in *Fouquet et al. (2009)*. The final pENTR4 construct contains 5499 bp RBP cDNA was recombined with pTW-Cherry gateway *Drosophila* transgenic vector. Transgenic flies were generated at Bestgene Inc., CA, USA and insertion was confirmed by genotyping.

## Generation of Aplip-WT1 and Aplip1-AxxA1 construct

To generate the cDNA of Aplip1 (with WT or mutated first PXXP motif), the full length cDNA clone of Aplip1 was kindly obtained from HYBRIGENICS Services, France and used as a template for cloning full length Aplip1 into pENTR/D-Topo (Invitrogen, Germany) using the following primers:
Aplip1-FL-FW 5′-CACCATGGCCGACAGCGAATTCGAGGAGTT-3′
Aplip1-FL-REV 5′-TCGGCGCGCCCACCCTTCTACTCAATGTAG-3′
   Through Gateway reaction, the construct was shuttled into GAL4/UAS vector and sent for injection at BestGene Inc., CA, USA. Point mutations were introduced into the constructs via mutated primers with the 'Quick Change II Site-Directed Mutagenesis Kit' from Stratagene, CA, USA. This induced a change of the prolines of PxxP1 (155-PEIP-160) into alanines (155-AEIA-160) after mutagenesis. Following primers were used:
Forward 5′ CGTCGTCGCAAGTTGGCGGAAATAGCGAAAAACAAGAAATCT 3′
Reverse 5′ AGATTTCTTGTTTTTCGCTATTTCCGCCAACTTGCGACGACG 3′

## Generation of peptides for crystallography

For crystallography constructs comprising either the RBP SH3-II (residue 1318–1382) or SH3-III (residue 1441–1507) domain of RBP were amplified by PCR and cloned into the pGEX-6P1 vector using *EcoRI* and *XhoI* restriction sites.
   The following primers were used:
SH3II_for 5′-CAGAATTCCGCTATTTTGTGGCCATGTTC-3′
SH3II_rev 5′-TACTCGAGTCACTCCACCTCGGAGACCAT-3′

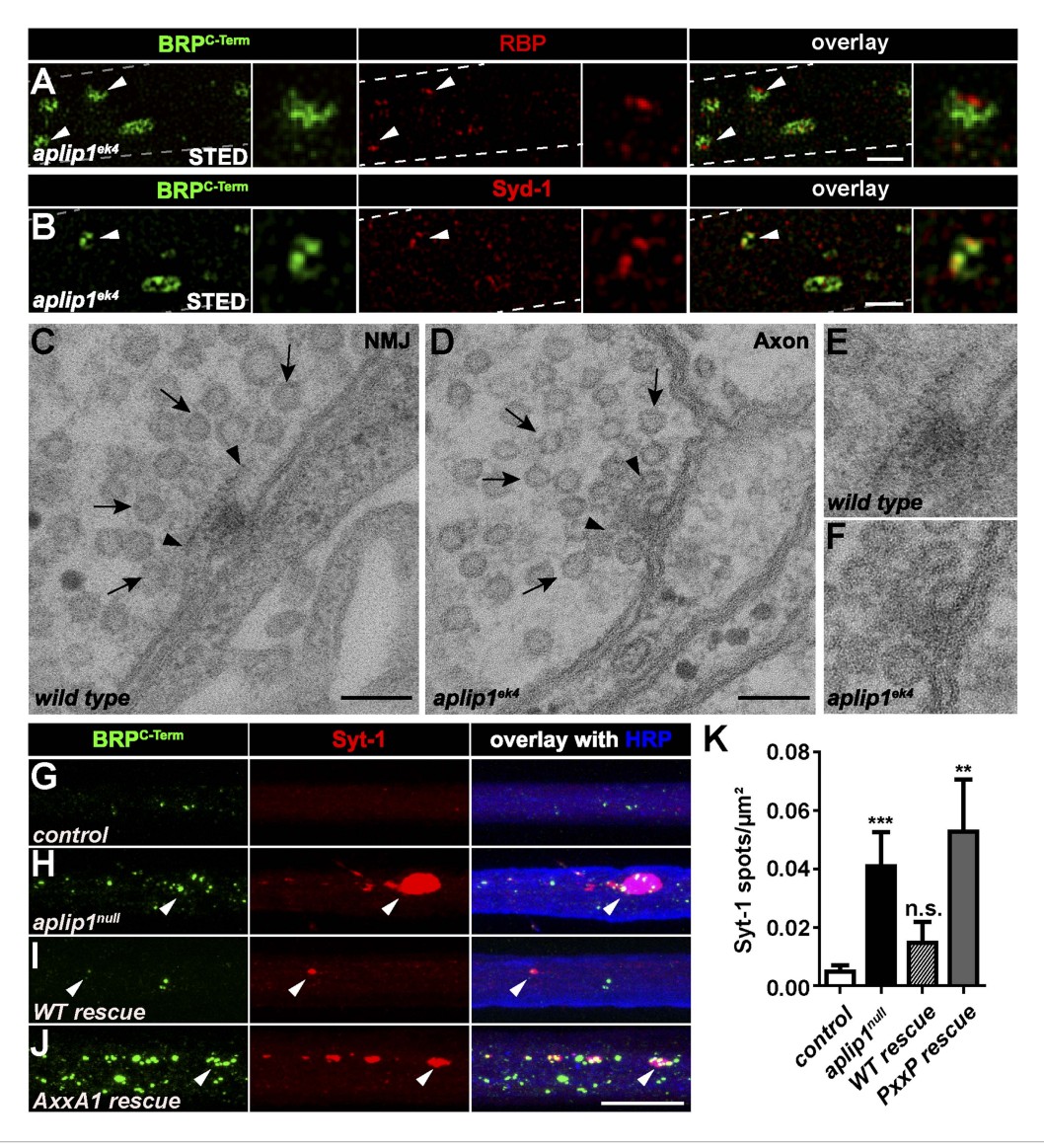

**Figure 7**. Ectopic AZ scaffold and synaptic vesicle (SV) accumulation in *aplip1* mutant axons. (**A**) Two-colour STED images of axonal aggregates in *aplip1ek4* mutants revealed that the structures observed (arrow heads) have identical BRP and RBP arrangement, as recently observed at presynaptic AZs (*Liu et al., 2011a*). Right panels display magnifications of single axonal AZ. Dashed lines indicate axonal plasma membrane. (**B**) Two-colour STED images of axonal aggregates in *aplip1ek4* mutants revealed that the structures observed (arrow head) have identical BRP and Syd-1 arrangement as observed at immature presynaptic AZs (*Owald et al., 2010*). Right panels display magnifications of single axonal AZ. Dashed lines indicate axonal plasma membrane. (**C**) Terminal T-bar (arrow heads) surrounded by SVs (arrows) taken from electron micrographs of WT third instar larvae after conventional embedding. (**D**) Ectopic axonal T-bar (arrow heads) taken from electron micrographs from *aplip1ek4* mutant third instar larvae after conventional embedding. SVs accumulate around the ectopic T-bar (arrows). (**E**) Magnification of (**C**). (**F**) Magnification of (**D**). (**G–J**) Nerve bundles of segment A1–A3 from third instar larvae of the genotypes indicated labeled with the Abs indicated. Syt-1 accumulates at a subset of axonal BRP aggregations in *aplip1null* and *AxxA1 rescue* (**H, J**) larvae, but not in control and WT rescue larvae (**G, I**). (**K**) Quantification of the number of Syt-1 spots per μm² axon. control (n = 12 nerves): 0.004 ± 0.002; *aplip1null* (n = 16 nerves): 0.040 ± 0.011; *WT rescue* (n = 13 nerves): 0.014 ± 0.007; *AxxA1 rescue* (n = 13 nerves): 0.052 ± 0.017. All panels show mean values and errors bars representing SEM. *p ≤ 0.05; **p ≤ 0.01; ***p ≤ 0.001; ns, not significant, p > 0.05, Mann–Whitney U test. Scale bars: (**A, B**) 500 nm; (**C, D**) 100 nm; (**G, J**) 10 μm.

The following figure supplement is available for figure 7:

**Figure supplement 1**. Ectopic AZ protein accumulations in motoneuronal driven Imac- and KHC-RNAi axons.

SH3III_for 5′-CAGAATTCAACATGCCCGTGAAGCGAATG-3′
SH3III_rev 5′-TACTCGAGTCAGTCCGCCAGGAAGTTAGA-3′

The resulting constructs comprise an N-terminal GST-tag that is followed by a PreScission cleavage site and the respective SH3 domain. Correctness of the DNA sequences was confirmed by DNA sequencing.

## Yeast two-Hybrid

The Yeast two-Hybrid screen for RBP interaction partners was carried out in collaboration with HYBRIGENICS Services, France using the LexA system (pB27 with bait; pP6 vector with prey) against the HYBRIGENICS *Drosophila melanogaster* head (adult) library. The vector maps of the bait and prey vectors are confidential (protected under material transfer agreement).

The plasmids (pP6 and pB27) encode tryptophan (Trp) and leucine (Leu) biosynthesis genes, and were successfully double transformed into the TATA strain lacking genes for synthesis of Leu and Trp which can be followed by positive growth in LT media. Reporter genes for the protein–protein interaction are HIS3, which can be later detected by growth on plates lacking histidin, as well as lacZ which allows the detection of interaction in a more quantitative fashion with a β-galactosidase assay. To transform the yeast cells with the pP6 and pB27 vector respectively the LiAc/single strand DNA/PEG technique was used (*Gietz and Schiestl, 2007*).

The RBP constructs for Y2H were cloned into pB27 bait vector. The RBP cDNA clone AT04807 (Drosophila Genomics Resource Centre, IN, USA) was used as a template for PCR reaction. For amplification the following primers were used:
5′-CAGAATTCGGTCAACCGGGACAACCGGGG-3′
5′-TAACTAGTTCAGTCGGGCGCGTCCGCCAGGA-3′

## Protein sequence of the bait fragment

### RBP SH3-II+III (length: 209 AA; Orientation C-term free [N-LexA-bait-C])

GQPGQPGQMPGAQKKPRYFVAMFDYDPSTMSPNPDGCDEELPFQEGDTIKVFGDKDADGFYWGELRG
RRGYVPHNMVSEVEDTTASMTAGGQMPGQMPGQMGQGQGVGVGGTAQVMPGQGAPQQSMRNVS
RDRWGDIYANMPVKRMIALYDYDPQELSPNVDAEQVELCFKTGEIILVYGDMDEDGFYMGELDGVRGLVP
SNFLADAPD

## Liquid Y2H ß-Galactosidase assay

The assay was carried out as described in JH Miller 'Experiments in Molecular Genetics' 1972 Cold Spring Harbor Laboratories pages 352–355.

The RBP constructs for Y2H were cloned into pB27 bait vector. The RBP cDNA clone AT04807 (Drosophila Genomics Resource Centre) was used as a template for PCR reaction. For amplification the following primers were used:

### RBP SH3-II
5′-CAGAATTCGGTCAACCGGGACAACCGGGG-3′
5′-TAACTAGTTCAGTCCTCCACCTCGGAGACC-3′

Giving rise to the following sequence

PGQPGQPGQMPGAQKKPRYFVAMFDYDPSTMSPNPDGCDEELPFQEGDTIKVFGDKDADGFYWGELR
GRRGYVPHNMVSEVED

### RBP SH3-III
5′-CAGAATTCATGCCCGTGAAGCGAATG-3′
5′-TAACTAGTTCAGTCGGGCGCGTCCGCCAGGA-3′

Giving rise to the following sequence

MPVKRMIALYDYDPQELSPNVDAEQVELCFKTGEIILVYGDMDEDGFYMGELDGVRGLVPSNFLADAPD

### RBP SH3-II+III
5′-CAGAATTCGGTCAACCGGGACAACCGGGG-3′
5′-TAACTAGTTCAGTCGGGCGCGTCCGCCAGGA-3′

## Giving rise to the following sequence

GQPGQPGQMPGAQKKPRYFVAMFDYDPSTMSPNPDGCDEELPFQEGDTIKVFGDKDADGFYWGELRG
RRGYVPHNMVSEVEDTTASMTAGGQMPGQMPGQMGQGQGVGVGGTAQVMPGQGAPQQSMRNVS
RDRWGDIYANMPVKRMIALYDYDPQELSPNVDAEQVELCFKTGEIILVYGDMDEDGFYMGELDGVRGLVPS
NFLADAPD

By applying the site-directed mutagenesis strategy, different constructs were designed for RBP using mutated primers. Mutagenesis was carried out by Dr Martin Meixner (SMB. GmbH, Germany,). The following point mutations were used:

RBP SH3-II*: Prolin$_{1373}$ → Leucin

## Giving rise to the following sequence

PGQPGQPGQMPGAQKKPRYFVAMFDYDPSTMSPNPDGCDEELPFQEGDTIKVFGDKDADGFYWGELR
GRRGYVLHNMVSEVED

RBP SH3-III*: Prolin$_{1500}$ → Leucin

## Giving rise to the following sequence

MPVKRMIALYDYDPQELSPNVDAEQVELCFKTGEIILVYGDMDEDGFYMGELDGVRGLVLSNFLADAPD

## The Aplip1 prey fragment only containing the first PXXP was generated from the full length fragment via PCR using the primers

Aplip first PXXP FW 5′-CGTACTCCATGGCTGAGGACGATGAGCTGGGCGA-3′
Aplip first PXXP REV 5′-CTGACTACTAGTTGGAGTCCTCGTCCATCAAGTA-3′

## Giving rise to the following sequence

### Aplip1-PXXP1 (length: 139 AA)

EDDELGDGLKVTLSSDGSLDTNDSFNSHRHHPLNHQDAIGGFLGMDTSGLGGNSAPVTIGASTDLLAPNT
AATRRRRKLPEIPKNKKSSILHLLGGSNFGSLADEFRNGGGGGGIPPAVRSGQQRSFLSLKCGYLMDEDS

The Aplip1 prey fragment only containing the second PXXP was generated from the full length fragment via PCR using the primers

Aplip second PXXP FW 5′-CGTACTCCATGGCTCTTCTAGGTGGCTCCAACTT-3′
Aplip second PXXP REV 5′-CTGACTACTAGTTCTGGCCAAAGGGCACGC-3′

## Giving rise to the following sequence

### Aplip1-PXXP2 (length: 100 AA)

LLGGSNFGSLADEFRNGGGGGGIPPAVRSGQQRSFLSLKCGYLMDEDSSPDSERMQSLGDVDSGHSTAHS
PNDFKSMSPQITSPVSQSPFPPPFGGVPFGQ

The Aplip1 prey fragment only containing the mutated first PXXP motif (AxxA) (see also Generation of Aplip-WT1 and Aplip1-AxxA1 construct) was generated from the full length fragment via PCR using the primers:

Forward 5′ CGTCGTCGCAAGTTGGCGGAAATAGCGAAAAACAAGAAATCT 3′
Reverse 5′ AGATTTCTTGTTTTTCGCTATTTCCGCCAACTTGCGACGACG 3′

## Giving rise to the following peptide sequence

### Aplip1-AXXA1 (length: 139 AA)

EDDELGDGLKVTLSSDGSLDTNDSFNSHRHHPLNHQDAIGGFLGMDTSGLGGNSAPVTIGASTDLLAPNT
AATRRRRKLAEIAKNKKSSILHLLGGSNFGSLADEFRNGGGGGGIPPAVRSGQQRSFLSLKCGYLMDEDS

The BRP constructs for Y2H were cloned into pB27 bait vector. Yeast two-hybrid constructs for BRP were obtained by PCR using the corresponding cDNA as template (modified from *Wagh et al., 2006*).

To generate BRP prey fragments the following primers were used:

Forward 5′ CAGCGGCCGCTCCAGTAACTAGCTCTGG 3′
Reverse 5′ TAACTAGTTTATATGTGCCGCTGGTAGTC 3′

Giving rise to the following peptide sequence

**BRP-D1 (length: 359 AA)**
PVTSSGVRSPGRVRRLQELPTVDRSPSRDYGAPRGSPLAMGSPYYRDMDEPTSPAGAGHHRSRSASRPPM
AHAMDYPRTRYQSLDRGGLVDPHDREFIPIREPRDRSRDRSLERGLYLEDELYGRSARQSPSAMGGYNTG
MGPTSDRAYLGDLQHQNTDLQRELGNLKRELELTNQKLGSSMHSIKTFWSPELKKERAPRKEESAKYSLIN
DQLKLLSTENQKQAMLVRQLEEELRLRMRQPNLEMRQQMEAIYAENDHLQREISILRETVKDLECRVETQK
QTLIARDESIKKLLEMLQAKGMGKEEERQMFQQMQAMAQKQLDEFRLEIQRRDQEILAMAAKMKTLEE
QHQDYQRHI
Forward 5′ CAGCGGCCGCGATGTTCCAGCAGATGC 3′
Reverse 5′ TAACTAGTTTACTGTGTGACTCTCAGCTCGGC 3′

Giving rise to the following peptide sequence

**BRP-D2 (length: 339 AA)**
MFQQMQAMAQKQLDEFRLEIQRRDQEILAMAAKMKTLEEQHQDYQRHIAVLKESLCAKEEHYNMLQTD
VEEMRARLEEKNRLIEKKTQGTLQTVQERNRLTSELTELKDHMDIKDRKISVLQRKIENLEDLLKEKDNQVDM
ARARLSAMQAHHSSSEGALTSLEEAIGDKEKQMAQLRDQRDRAEHEKQEERDLHEREVADYKIKLRAAESE
VEKLQTRPERAVTERERLEIKLEASQSELGKSKAELEKATCEMGRSSADWESTKQRTARLELENERLKHDLER
SQNVQKLMFETGKISTTFGRTTMTTSQELDRAQERADKASAELRRTQAELRVTQ
Forward 5′ CAGAATTCGAGCGGGCCGACAAGGC 3′
Reverse 5′ TAACTAGTTCACATTTGCGCCTTCTC 3′

Giving rise to the following peptide sequence

**BRP-D3 (length: 636 AA)**
ERADKASAELRRTQAELRVTQSDAERAREEAAALQEKLEKSQGEVYRLKAKLENAQGEQESLRQELEKAQ
SGVSRIHADRDRAFSEVEKIKEEMERTQATLGKSQLQHEKLQNSLDKAQNEVDHLQDKLDKACTENRRLV
LEKEKLTYDYDNLQSQLDKALGQAARMQKERETLSLDTDRIREKLEKTQVQLGRIQKERDQFSDELETLKER
SESAQTLLMKAARDREAMQTDLEVLKERYEKSHAIQQKLQMERDDAVTEVEILKEKLDKALYASQKLIDEK
DTSNKEFEKMLEKYDRAQNEIYRLQSRCDTAEADRARLEVEAERSGLAASKAREDLRKLQDESTRLQEACD
RAALQLSRAKECEDNARSELEHSRDRFDKLQTDIRRAQGEKEHFQSELERVTYELERAHAAQTKASASVEA
AKEEAAHYAVELEKMRDRYEKSQVELRKLQDTDTFGRETRRLKEENERLREKLDKTLMELETIRGKSQYESE
SFEKYKDKYEKIEMEVQNMESKLHETSLQLELSKGEVAKMLANQEKQRSELERAHIEREKARDKHEKLLKEV
DRLRLQQSSVSPGDPVRASTSSSSALSAGERQEIDRLRDRLEKALQSRDATELEAGRLAKELEKAQM
Forward 5′ CAGCGGCCGCCCTGCAACAGTCCTCGG 3′
Reverse 5′ TAACTAGTTTACAACTCTGTGACCAG 3′

Giving rise to the following peptide sequence

**BRP-D4 N-term (length: 348 AA)**
LQQSSVSPGDPVRASTSSSSALSAGERQEIDRLRDRLEKALQSRDATELEAGRLAKELEKAQMHLAKQQEN
TESTRIEFERMGAELGRLHDRLEKAEAEREALRQANRSGGAGAAPHPQLEKHVQKLESDVKQLAMEREQL
VLQLEKSQEILMNFQKELQNAEAELQKTREENRKLRNGHQVPPVAAPPAGPSPAEFQAMQKEIQTLQQK
LQESERALQAAGPQQAQAAAAAGASREEIEQWRKVIEQEKSRADMADKAAQEMHKRIQLMDQHIKDQ
HAQMQKMQQQMQQQQQAAQQAVQQAAQQQQSAAGAGGADPKELEKVRGELQAACTERDRFQQ
QLELLVTEL
Forward 5′ CAGAATTCAAGAGCAAGATGTCCAAC 3′
Reverse 5′ TAACTAGTTTAGAAAAAGCTCTTCAA 3′

Giving rise to the following peptide sequence

**BRP-D4 C-term (length: 227 AA)**
SKMSNQEQAKQLQTAQQQVQQLQQQVQQLQQQMQQLQQAASAGAGATDVQRQQLEQQQKQLE
EVRKQIDNQAKATEGERKIIDEQRKQIDAKRKDIEEKEKKMAEFDVQLRKRKEQMDQLEKSLQTQGGGAA
AAGELNKKLMDTQRQLEACVKELQNTKEEHKKAATETERLLQLVQMSQEEQNAKEKTIMDLQQALKIAQ
AKVKQAQTQQQQQQDAGPAGFLKSFF

## IP

IP of elav-Gal4/+;UAS-Aplip1$^{GFP}$/+ was performed as described in *Depner et al. (2014)*. In brief, the experiment was performed as following, 500 µl adult fly heads were mechanically homogenized in 500 µl lysis buffer (50 mM Tris pH 8.0, 150 mM KCl, 1 mM MgCl$_2$, 1 mM EGTA, 10% glycerol containing protease inhibitor cocktail [Roche, Germany]). 0.4% Sodium deoxycholate was added, and the lysate was incubated on ice for 30 min. The lysate was diluted 1:1 with sodiumdeocycholat-free lysis buffer, then 1% Triton X-100 was added and lysate was kept on the wheel at 4°C for 30 min. After centrifugation for 15 min at 16,000×g, the supernatant was used in IPs with GFP-Trap-A beads and blocked agarose beads as binding control (Chromotek, Germany). After incubation overnight at 4°C, beads were washed in buffer without detergent and glycerol. Proteins were eluted from the beads with SDS sample buffer. Afterward, the SDS-PAGE samples were subjected to Western blot (WB).

## SDS-PAGE and Tris-Acetate gel electrophoresis

The gel electrophoresis for both SDS-PAGE and Tris-acetate gels was conducted according to the standard protocols (*Laemmli, 1970*; *Schägger, 2006*). Colloidal Coomassie blue stain was used to detect proteins based on manufacture protocol (Carl-Roth, Germany and Invitrogen). For BRP, RBP and Aplip1, standard SDS-PAGE gels (6–12%) were used to separate the target protein.

## WB analysis

Following the separation by gel electrophoresis, the proteins were transferred into a nitrocellulose membrane by wet transfer procedure using cold transfer buffer (25 mM Tris, pH 8.0, 150 mM glycine, 20% methanol). For visualization of proteins, the membrane was stained using Ponceau-S staining solution (Sigma–Aldrich, MO, USA). 5% milk powder in phosphate buffered saline (PBS) was used for blocking of the membrane. Following the blocking, the membrane was incubated with the primary Abs guinea pig BRP$^{Last200}$ (1:5000, Ullrich et al., in submission) and rabbit RBP$^{SH3-II+III}$ (1:1000, *Depner et al., 2014*) at 4°C for overnight. After several washing steps, the membrane was incubated with horseradish peroxidase (HRP) conjugated secondary Abs (Dianova, Germany). For detection, an enhanced chemoluminescence substrate (GE Healthcare, United Kingdom) was used and the X-ray film (GE Healthcare) development was carried manually.

## Immunostaining

Larval filets were dissected and stained as described previously (*Owald et al., 2010*). The following primary Abs were used: rabbit BRP$^{N-term}$ (1:500; *Qin et al., 2005*); rabbit Liprin-α (1:500; *Fouquet et al., 2009*); rabbit Syd-1 (1:500; *Owald et al., 2010*); rabbit Rab3 (1:500; *Graf et al., 2009*); rabbit RBP$^{C-term}$, rabbit RBP$^{SH3-II+III}$ (1:500; *Depner et al., 2014*); rabbit Syt1-CL1 (1:1000; gift from N Reist [*Mackler et al., 2002*], Colorado State University, CO, USA); mouse GFP (3E6) (1:500, Life Technologies, Germany), mouse Nc82 = anti-BRP$^{C-term}$ (1:100, Developmental Studies Hybridoma Bank, University of Iowa, Iowa City, IA, USA). Except for staining against Cac$^{GFP}$, where larvae were fixed for 5 min with ice-cold methanol, all fixations were performed for 10 min with 4% paraformaldehyde in 0.1 mM PBS. Secondary Abs for standard immunostainings were used in the following concentrations: goat anti-HRP-Cy5 (1:250, Jackson ImmunoResearch, PA, USA); goat anti-rabbit Cy3 (1:500, Life Technologies); goat anti-mouse Alexa-Fluor-488 (1:500, Life Technologies). Larvae were mounted in vectashield (Vector labs, United Kingdom). Secondary Abs for STED were used in the following concentrations: For *Figures 1H, 7A*: goat anti-mouse Atto594 (1:250); goat anti-rabbit Atto594 (1:250); goat anti-mouse Atto647N (1:100), goat anti-rabbit Atto647N (1:100) (ATTO-TEC, Germany). For *Figure 7B*: goat anti-mouse Atto590 (1:100); goat anti-rabbit star635 1:100 (Atto590 [ATTO-TEC] and star635 [Abberior, Germany]) coupled to respective IgGs (Dianova, Germany). For *Figure 7—figure supplement 1A–C*: goat anti-mouse Alexa-Fluor-488 (1:500, Life Technologies) and goat anti-rabbit Alexa-Fluor-532 (1:500, Life Technologies) was used. For STED imaging larvae were mounted in Mowiol (Max-Planck Institut for Biophysical Chemistry, Group of Stefan Hell) or Prolong Gold antifade reagent (Life Technologies; *Figure 7—figure supplement 1A–C*).

## Image acquisition, processing and analysis

Confocal microscopy was performed with a Leica TCS SP5 (all except for *Figure 4G–J* and *Figure 7G–J*) or a Leica SP8 (*Figure 4G–J* and *Figure 7G–J*) confocal microscope (Leica Microsystems, Germany).

STED microscopy was performed with a custom-built STED-microscope (see below). Images of fixed and live samples were acquired at room temperature. Confocal imaging of axons was done using a z step of 0.25 µm. The following objective was used: 63× 1.4 NA oil immersion for NMJ confocal imaging. All confocal images were acquired using the LCS AF software (Leica, Germany). Images from fixed samples were taken from third instar larval nerve bundles (segments A1–A3). Images for figures were processed with ImageJ software to enhance brightness using the brightness/contrast function. If necessary images were smoothened (0.5–1 pixel Sigma radius) using the Gaussian blur function.

Quantifications of axonal spot number and size were performed following an adjusted manual (*Andlauer and Sigrist, 2012*), briefly as follows. The signal of a HRP-Cy5 Ab was used as template for a mask, restricting the quantified area to the shape of the axon/nerve bundles. The original confocal stacks were converted to maximal projections and after background subtraction, a mask of the axonal area was created by applying a certain threshold to remove the irrelevant lower intensity pixels. The segmentation of single spots was done semi-automatically via the command 'Find Maxima' and by hand with the pencil tool and a line thickness of 1 pixel. To remove high frequency noise a Gaussian blur filter (0.5 pixel Sigma radius) was applied. The processed picture was then transformed into a binary mask using the same lower threshold value as in the first step. This binary mask was then projected onto the original unmodified image using the 'min' operation from the ImageJ image calculator. The axonal spots of the resulting images were counted with the help of the 'analyze particle' function with a lower threshold set to 1. The spot density was obtained by normalizing the total number of analyzed particles to the axonal area measured via HRP. Colocalization of RBP/BRP spots (*Figure 1G*) was counted manually.

Data were analyzed using the Mann–Whitney U test for linear independent data groups. Means are annotated ±SEM. Asterisks are used to denote significance: $*p < 0.05$; $**p < 0.01$; $***p < 0.001$; n.s. (not significant), $p > 0.05$.

## STED microscopy

For *Figures 1H, 7A* two-colour STED images were recorded with a custom-built STED microscope which combines two pairs of excitation and STED laser beams all derived from a single supercontinuum laser source (*Bückers et al., 2011*). For *Figure 7B* STED microscopy was performed as previously described in *Li et al. (2014)*. Here, two-colour STED images were recorded on a custom-built STED-microscope (*Göttfert et al., 2013*), which combines two pairs of excitation laser beams of 595 nm and 640 nm wavelength with one STED fiber laser beam at 775 nm. All STED images were acquired using Imspector Software (Max Planck Innovation GmbH). STED images were processed using a linear deconvolution function integrated into Imspector Software (Max Planck Innovation GmbH, Germany). Regularization parameters ranged from $1e^{-09}$ to $1e^{-10}$. The point spread function (PSF) for deconvolution was generated by using a 2D Lorentz function with its half-width and half-length fitted to the half-width and half-length of each individual image. For *Figure 7—figure supplement 1*, STED microscopy was performed with a Leica TCS SP5 time gated STED microscope equipped with a 100× 1.4 NA objective using the LCS AF software (Leica) for image *acquisition*. Alexa-Fluor-488 and Alexa-Fluor-532 were excited using a pulsed white light laser at 488 and 545 nm, respectively. STED was achieved with a continous STED laser at 592 nm. In gSTED mode time gated detection started at 1.2 ns–6 ns for Alexa488 while for Alexa532 gating time was set to 2.3 ns–6 ns. Raw gSTED images were deconvolved using the built-in algorithm of the LAS AF software (Signal intensity; regularisation parameter 0.05). The PSF was generated using a 2D Lorentz function with the full-width half maximum set to 60 nm. Images for figures were processed with ImageJ software to remove obvious background, enhance brightness/contrast and smoothened (1 pixel Sigma radius) using the Gaussian blur function.

## Live imaging and analysis

Live imaging was performed as previously described (*Füger et al., 2007*). Briefly, third instar larvae were put into a live imaging chamber and anaesthetized with 10–20 short pulses of a desflurane-air mixture until the heartbeat completely stopped. For assessing axonal transport, axons immediately after exiting the ventral nerve cord were imaged for 10 min using timelapse confocal microscopy. The flux was determined by manually counting the number of moving spots (unidirectional for >3

frames) passing a virtual line in the middle of the nerve bundle. Mean flux was calculated by pooling results from at least three independent larvae and at least six nerves. If little or no flux was observed, additional nerves were imaged to avoid any bias from selecting specific nerves.

## ITC

ITC experiments were performed at 25°C on an iTC200 microcalorimeter (Malvern Instruments Ltd., United Kingdom). The same peptides were employed as used for the co-crystallization experiments (see below). Lyophilized peptides were resuspended in the final buffer of the proteins (10 mM Tris-HCl pH 7.4, 100 mM NaCl). RBP SH3-II and SH3-III were both provided at a concentration of 150 µM, RBP SH3-II+III was provided at 78 µM. The proteins were titrated with 16 injections of 2.5 µl of either Aplip1, Cac, RIM1 or RIM2 peptide at a concentration of 2 mM with 2-min intervals. The released heat was obtained by integrating the calorimetric output curves. Binding parameters were calculated using the Origin5 software using the 'One Set of Sites' curve fitting model provided by the software.

### The following peptides were used

#### RBP SH3-I

RFPYDPPEEAEGELSLCAGDYLLVWTSGEPQGGYLDAELLDGRRGLVPASFVQRLVG

#### RBP SH3-II

RYFVAMFDYDPSTMSPNPDGCDEELPFQEGDTIKVFGDKDADGFYWGELRGRRGYVPHNMVSEVE

#### RBP SH3-III

KRMIALYDYDPQELSPNVDAEQVELCFKTGEIILVYGDMDEDGFYMGELDGVRGLVPSNFLAD

#### RBP SH3-II+III

RYFVAMFDYDPSTMSPNPDGCDEELPFQEGDTIKVFGDKDADGFYWGELRGRRGYVPHNMVSEVEDTT
ASMTAGGQMPGQMPGQMGQGQGVGVGGTAQVMPGQGAPQHSMRNVSRDRWGDIYANMPVKRM
IALYDYDPQELSPNVDAEQVELCFKTGEIILVYGDMDEDGFYMGELDGVRGLVPSNFLAD
Aplip-PxxP1: TRRRRKLPEIPKNKK
Cac: IGRRLPPTPSKPSTL
RIM1: GRQLPQVPVRSG
RIM2: GRQLPQLPPKGT

## Protein expression and purification for crystallization

Protein expression was conducted using chemically competent *Escherichia coli* BL21-CodonPlus-RIL cells. The cells were grown in autoinduction ZY-medium (*Studier, 2005*) with ampicillin and chloramphenicol for 4 hr at 37°C. Afterwards, the temperature was decreased to 18°C, and the cells were grown overnight. The cells were harvested by centrifugation at 8,000×*g* for 6 min. The cell pellet was resuspended in resuspension buffer (40 mM Tris/HCl pH 7.5 at RT, 250 mM NaCl, 1 mM DTT, 10 mg/l lysozyme and 5 mg/l DNase I) and subsequently lysed by sonication for 20 min. The lysate was centrifuged at 56,000×*g* for 45 min to pellet the cell debris. The supernatant was applied for affinity chromatography using 10 ml glutathione sepharose 4B (GE Healthcare). Hereafter, two washing steps were performed using 80 ml washing buffer (20 mM Tris/HCl pH 7.5 at RT, 250 mM NaCl, 1 mM DTT) for each step. The GST-tag of the respective SH3 domain was cleaved off on the beads using PreScission protease (1 mg/ml). Therefore 40 ml washing buffer with PreScission protease in a molar ratio of 1:30 to the maximum loading capacity of the glutathione sepharose were incubated with the beads at 4°C while gently rotating overnight. The PreScission-cleaved constructs were purified using a Superdex 75 26/60 column (GE Healthcare). The protein containing fractions were pooled and concentrated using a 3 kDa molecular weight cut-off concentrator (Millipore, Germany). Protein concentrations were determined by UV-absorption.

## Crystallization and crystal cooling

For crystallization experiment the RBP SH3-II was concentrated to 56 mg/ml and the RBP SH3-III to 62 mg/ml. The same peptides as for ITC measurements were used and synthesized at the Leibniz Institute for Molecular Pharmacology with N-terminal acetylation and C-terminal amidation. The

unsolubilized peptides were mixed in a fivefold molar excess with the protein solution and incubated for 2 hr on ice. Insoluble peptide was removed by centrifugation (16,000×g for 1 min) prior to crystallization experiments. All crystallization experiments were carried out at 291 K in a sitting drop setup. Crystals of RBP SH3-II bound to the Aplip1 peptide were obtained over a reservoir solution composed of 2.2–2.6 M ammonium sulfate, 0.1 M bicine with final pH 9. For cryoprotection, the crystals were transferred to a reservoir solution supplemented with 25% (vol/vol) glycerol. Crystals of RBP SH3-II bound to Cac were obtained over a reservoir solution of 0.2 M Ca(Ac)$_2$, 0.1 M MES pH 6.0, and 20% (wt/vol) polyethylenglycol (PEG) 8000. For cryoprotection, the crystals were transferred to a reservoir solution supplemented with 15% (vol/vol) PEG 400. Crystals of RBP SH3-III bound to the Cac peptide appeared over a reservoir solution of 0.2 M Li$_2$SO$_4$, 0.1 M MES pH 6.5, and 30% (vol/vol) PEG 400. After cryoprotection the crystals were flash-cooled in liquid nitrogen.

## Diffraction data collection and analysis as well as structure determination

Synchrotron diffraction data were collected at the beamline 14.2 of the MX Joint Berlin laboratory at BESSY (Berlin, Germany). X-ray data collection was performed at 100 K. Diffraction data were processed with the XDS package (*Kabsch, 2010*). The diffraction data of RBP SH3-II/Aplip1-PxxP1 were initially indexed in *P*622. Cumulative intensity distribution analysis as well as calculation of the moment of the observed intensity/amplitude distribution performed with PHENIX.XTRIAGE and POINTLESS (*Evans, 2011*) indicated an unusual intensity distribution, likely caused by twinning. For determination of the correct space group, the diffraction data were processed in *P*1. Subsequently, the structure was solved by molecular replacement with the program PHASER (*McCoy et al., 2007*). We used the NMR structure of the SH3-II domain of human RBP (PDB entry 2CSQ) as search model and could locate 24 copies of the SH3 domain. Next the diffraction data and the coordinates of our molecular replacement were analysed by the program ZANUDA (*Lebedev and Isupov, 2014*) revealing that sixfold is in fact broken and *C*2 is the true symmetry, with sixfold twinning with the six twin operators: h, h, l; h, −k, −l; 1/2h − 3/2k, −1/2h − 1/2k, −k; −1/2h + 3/2k, 1/2h + 1/2k, −l; −1/2h − 3/2k, −1/2h + 1/2k, −l and 1/2h + 3/2k, 1/2h − 1/2k, −l. In total we could locate in the asymmetric unit 12 copies of RBP SH3-II bound to Aplip1-PxxP1. The crystals of RBP SH3-II and SH3-III bound to the Cac peptide have *P*2$_1$ and *I*222 symmetry, respectively. Analyses of the diffraction data of the complex of RBP SH3-II and Cac revealed one pseudo-merohedral twin operator (h, −k, −h − l), that was later included in the refinement protocol. The structures of RBP SH3-II and SH3-III each bound to the Cac derived peptide were solved by molecular replacement with our previously determined structure of RBP SH3-II. The asymmetric unit of RBP SH3-II bound to Cac contains 10 complexes and of RBP SH3-III bound to Cac one complex, respectively.

## Refinement and validation

The refined molecular replacement solution clearly revealed the presence of the bound Aplip1-PxxP1 peptide in 2m*Fo* − D*Fc* and m*Fo* − D*Fc* electron density maps. For refinement, a set of 4.7% of *R*$_{free}$ reflections was generated in *P*622 and then expanded to *C*2 to insure equal distribution of the *R*$_{free}$ reflections in all six twin domains. For calculation of the free R-factor of the other two data sets, a randomly generated set of 5% of the reflections from the diffraction data set was used and excluded from the refinement. The structure was manually built in COOT (*Emsley et al., 2010*) and refined in REFMAC 5.8.0073 (*Murshudov et al., 2011*) with intensity based twin refinement. In final stages TLS refinement was applied with every protein and peptide chain as single TLS group. The structures with bound Cac peptide were refined with PHENIX.REFINE (*Adams et al., 2010*; *Afonine et al., 2012*). Water molecules were picked with COOT and manually inspected. All structures were evaluated with MOLPROBITY (*Chen et al., 2010*) and PROCHECK (*Laskowski et al., 1993*). Figures were drawn with PYMOL (*DeLano, 2002*).

## EM

Conventional embedding was performed as described previously (*Fouquet et al., 2009*).

## Statistics

Data were analyzed using the Mann–Whitney rank sum test for linear independent data groups (Prism; GraphPad Software, Inc.). Means are annotated ± SEM. Asterisks are used to denote significance (*p < 0.05; **p < 0.01; ***p < 0.005; not significant, p > 0.05).

## Acknowledgements

This work was supported by grants from the Deutsche Forschungsgemeinschaft to SJS (Exc 257, TP A3/SFB 958, TP B9/SFB665) and JHD, MCW, BL, SJS (TP A6/SFB 958). MS was supported by a Ph.D. fellowship from the Max Delbrück Center for Molecular Medicine and a Boehringer Ingelheim Fonds Ph.D. fellowship. MB was supported by a Ph.D. fellowship from the graduate school GRK 1123 funded by the DFG. UR was supported by the International Max Planck Research School (IMPRS) on Multiscale Bio-Systems. We are grateful to Noreen Reist for generously contributing antibodies. We would like to thank Janine Lützkendorf (SFB958) for help with genetic analysis, Madeleine Brünner and Anastasia Stawrakakis for excellent technical assistance and Astrid Petzoldt for comments on the manuscript. We thank Thorsten Mielke for help with electron microscopy. We are grateful to G Bourenkov for discussion on the unusual twinning phenomenon. We accessed beamlines of the BESSY II (Berliner Elektronenspeicherring-Gesellschaft für Synchrotronstrahlung II) storage ring (Berlin, Germany) via the Joint Berlin MX-Laboratory sponsored by the Helmholtz Zentrum Berlin für Materialien und Energie, the Freie Universität Berlin, die Humboldt-Universität zu Berlin, the Max-Delbrück Centrum, and the Leibniz-Institut für Molekulare Pharmakologie. We thank C Weise SFB958/Z3 for mass spectrometric analysis.

## Additional information

### Funding

| Funder | Grant reference | Author |
|---|---|---|
| Deutsche Forschungsgemeinschaft (DFG) | SFB958/A3 | Markus C Wahl, Bernhard Loll, Stephan J Sigrist |
| Deutsche Forschungsgemeinschaft (DFG) | SFB665/B9 | Stephan J Sigrist |
| Deutsche Forschungsgemeinschaft (DFG) | Graduate Student Fellowship, GRK 1123 | Mathias A Böhme |
| Deutsche Forschungsgemeinschaft (DFG) | SFB958/Z3 | Markus C Wahl, Bernhard Loll |
| Deutsche Forschungsgemeinschaft (DFG) | SFB958/A6 | Markus C Wahl, Bernhard Loll, Stephan J Sigrist |
| Max-Planck-Gesellschaft | International Max Planck Research School | Ulises Rey |

The funders had no role in study design, data collection and interpretation, or the decision to submit the work for publication.

### Author contributions

MS, MAB, BL, Conception and design, Acquisition of data, Analysis and interpretation of data, Drafting or revising the article; JHD, Conception and design, Acquisition of data, Analysis and interpretation of data; HB, MMM, UR, NR, TM, NH, SR-A, FG, DK, CQ, TFMA, SWH, Acquisition of data, Analysis and interpretation of data; SK, Acquisition of data, Contributed unpublished essential data or reagents; CAC, Conception and design, Contributed unpublished essential data or reagents; MCW, Conception and design, Drafting or revising the article; SJS, Conception and design, Analysis and interpretation of data, Drafting or revising the article

### Author ORCIDs

Matthias Siebert, http://orcid.org/0000-0002-1739-5825
Mathias A Böhme, http://orcid.org/0000-0002-0947-9172
Niraja Ramesh, http://orcid.org/0000-0002-2867-7131
Till FM Andlauer, http://orcid.org/0000-0002-2917-5889
Bernhard Loll, http://orcid.org/0000-0001-7928-4488

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
