## [Decision Letter]

Thank you for sending your work entitled “A high affine RIM-binding protein/Aplip1 interaction prevents the formation of ectopic axonal active zones” for consideration at *eLife*. Your article has been favorably evaluated by K VijayRaghavan (Senior editor) and two reviewers, one of whom, Mani Ramaswami has agreed to share his identity.

The Reviewing editor and the other reviewer discussed their comments before we reached this decision, and the Reviewing editor has assembled the following comments to help you prepare a revised submission.

In this manuscript, Siebert et al. use the fly larval NMJ system to study the important but poorly understood question of active zone transport. The data are very interesting and the topic is important for understanding synapse development and axonal trafficking.

Observations presented show that BRP and RBP are normally co-transported in motor axons of *Drosophila*. RBP binds the transport adaptor protein Aplip1/JIP via its C-terminal SH3 domain, which interacts with a proline-rich (PxxP) motif of Aplip1/JIP1. Point mutations in this PxxP motif perturb normal transport of RBP and BRP and result in ectopic AZ- like structures forming in axonal membranes. Aplip1 also binds BRP directly, which promotes BRP transport independently of RBP. The imaging and localization data are exceptionally nice, the biochemical studies are supported by X-ray crystallography, and the mutations analyzed appear as precise as one could hope for. Thus, the analysis is rigorous and the data of high quality.

Substantive concerns:

1) The authors conclude that transport adaptors bind and sequester AZ proteins in an inactive state, preventing AZ assembly until the proteins have reached the presynaptic terminal. However, this is only one possible interpretation and at least one additional experiment (of a few suggested below) will be necessary to support or qualify this conclusion.

There are two models to explain the ectopic synapse formation in Aplip1 mutant (the EM pictures show nicely that the ectopic accumulations are well organized AZs). Model 1: Aplip1 binds to both AZ proteins (maybe more AZ proteins) and acts as chaperons to prevent critical binding interactions between the AZ proteins that drive synapse formation. Model 2: Aplip1 acts as an adaptor protein for motor proteins to transport these AZ proteins. The trafficking of AZ complexes naturally antagonizes their ability to assemble into T-bars. The authors clearly favor the first model. The second model is also possible if not likely because the authors do see a dramatic trafficking (movement phenotype) in the *aplip1* mutant. One way to test these models is to examine the motor mutants for similar phenotypes, and/or to downregulate motor proteins in the *aplip1* mutant background to see if that can further strengthen the phenotype. If the motor mutants phenocopy *aplip1* and enhances *aplip1*, then the second model may appear more likely.

2) What is the relationship between srpk79D and Aplip1?

The evidence BRP and RBP trafficking are independent of each other is good, yet they are also co-trafficked. This seems to argue that BRP and RBP are on different packets, these packets are moving together. A critical question is whether Aplip1 also binds to BRP with the same motif as it uses to bind to RBP. Their genetic rescue results seem to argue this is the case because the AxxA1 mutant form of Aplip1 could not even rescue the BRP accumulation. On the other hand, it seems to be surprising that such specific mutation disrupt the binding to a coiled coil domain of BRP. It will help for the authors to characterize the binding between BRP and Aplip more carefully.

3) Is there an estimate of how many axons are being recorded for trafficking events at the same time? This helps to understand the frequency of the movements better.

4) What is the size of these ectopic puncta compared with synaptic BRP puncta at the active zone? How about the size of the moving puncta vs stationary puncta in the axon?

---

## [Author Response]

*1) The authors conclude that transport adaptors bind and sequester AZ proteins in an inactive state, preventing AZ assembly until the proteins have reached the presynaptic terminal. However, this is only one possible interpretation and at least one additional experiment (of a few suggested below) will be necessary to support or qualify this conclusion*.

*There are two models to explain the ectopic synapse formation in Aplip1 mutant (the EM pictures show nicely that the ectopic accumulations are well organized AZs). Model 1: Aplip1 binds to both AZ proteins (maybe more AZ proteins) and acts as chaperons to prevent critical binding interactions between the AZ proteins that drive synapse formation. Model 2: Aplip1 acts as an adaptor protein for motor proteins to transport these AZ proteins. The trafficking of AZ complexes naturally antagonizes their ability to assemble into T-bars. The authors clearly favor the first model. The second model is also possible if not likely because the authors do see a dramatic trafficking (movement phenotype) in the* aplip1 *mutant. One way to test these models is to examine the motor mutants for similar phenotypes, and/or to downregulate motor proteins in the* aplip1 *mutant background to see if that* can *further strengthen the phenotype. If the motor mutants phenocopy* aplip1 *and enhances* aplip1*, then the second model may appear more likely*.

We fully agree with the reviewers that further investigating these mechanistic interpretations, which obviously are not mutually exclusive, is interesting and relevant. Thus, we now performed two-colour STED immuno-stainings of *ctrl, OK6::UAS-Imac-RNAi* and *OK6::UAS-KHC-RNAi* stained for BRP and RBP (Figure 7—figure supplement 1). Additionally, we preformed systematic EM on axonal sections of these genotypes (10–15 axonal cross sections per genotype).

In *aplip1* mutants, we found BRP/RBP accumulations facing the axonal membrane (“T-bars”), clearly resembling synaptic AZs. After depriving motoneurons of Imac (*Ok6::UAS-Imac-RNAi*), BRP/RBP accumulations were rather scarce, although clearly more than in controls (*Ok6::+*). Despite extensive EM analysis, not a single T-bar could be observed after depriving Imac as well as in controls.

In *khc* mutants (*Ok6::UAS-KHC-RNAi*), severe axonal RBP/BRP accumulations were observed. However, STED analysis did not recover any preferential orientation of this ectopic BRP/RBP material towards the axonal plasma membrane and a generally less organized appearance than in *aplip1* mutants. Motoneuronal downregulation of KHC leads to severe accumulations of mitochondria and irregular shaped SVs as seen via EM (data not shown). But despite intense investigation, we found just one axonal structure that might represent a T-bar-like structure in *khc* mutants (Figure 7—figure supplement 1, magnification in E, F). In contrast, proper T-bars were identified in *aplip1* mutant axons with ease. Therefore while effective trafficking of AZ complexes might play a role in suppressing premature T-bar assembly, our data still are fully consistent with Aplip1 binding to AZ proteins being of particular importance for suppressing premature T-bar assembly.

We have now added the following to the Results of our manuscript: “To further support the importance of adaptor protein – cargo interaction in blocking ectopic AZ assembly we downregulated the expression of motor proteins. This also leads to transport defects and ectopic axonal AZ protein accumulations but in principle leaving the adaptor protein – cargo interaction intact. Interestingly, motoneuronal driven Imac-RNAi led to only few axonal BRP/RBP accumulations although with no preference concerning their direction in relation to the axonal plasma membrane (Figure 7—figure supplement 1; arrow heads). In contrast motoneuronal driven KHC-RNAi showed prominent axonal aggregates consistent of BRP/RBP but most of the time showing an irregular, elongated shape (Figure 7—figure supplement 1; arrow heads). As mentioned above, proper T-bars were identified in *aplip1* mutant axons with ease. In contrast, systematic EM analysis of *khc* mutant axons revealed just one electron dense material that showed a T-bar-like appearance (Figure 7—figure supplement 1; arrow head, magnifications in E, F) but never in ctrl or motoneuronal driven Imac-RNAi.”

Furthermore, we have now added the following sentences to the Discussion: “The down-regulation of the motor protein KHC also provoked severe axonal co-accumulations of BRP and RBP but per se should leave the adaptor protein-AZ cargo interaction intact. In contrast to *aplip1,* the axonal aggregations in *khc* mutants adapted irregular shapes most of the time, likely not representing T-bar-like structures. Thus, our data suggest a mechanistic difference when comparing the consequences between eliminating adaptor cargo interactions with a direct impairment of motor functions. Still, we cannot exclude that trafficking of AZ complexes naturally antagonizes their ability to assemble into T-bars.”

2) What is the relationship between srpk79D and Aplip1?

The evidence BRP and RBP trafficking are independent of each other is good, yet they are also co-trafficked. This seems to argue that BRP and RBP are on different packets, these packets are moving together. A critical question is whether Aplip1 also binds to BRP with the same motif as it uses to bind to RBP. Their genetic rescue results seem to argue this is the case because the AxxA1 mutant form of Aplip1 could not even rescue the BRP accumulation. On the other hand, it seems to be surprising that such specific mutation disrupt the binding to a coiled coil domain of BRP. It will help for the authors to characterize the binding between BRP and Aplip more carefully.

Thank you very much for pointing our attention to the yeast-two-hybrid data concerning a putative interaction between Bruchpilot and Aplip1. We sought to characterize the binding between BRP and Aplip1 more in detail and tested the interaction of several different Aplip1 constructs to several BRP constructs. Despite substantial efforts, however, we experienced a lack of consistency in the results and therefore are not convinced by the robust character of this interaction. We thus refrain from this statement, removed the Aplip1::BRP interaction part of Figure 5, and reformulated the relevant passages of the manuscript accordingly. To reconfirm all other yeast-two-hybrid defined Aplip1 interactions we describe in the present study, we re-tested all of them. We could reconfirm all of them (Figure 3).

As our manuscript mainly focuses on the high affinity interaction of RBP and Aplip1 and specifically on its influence on AZ protein transport, we think that the principal message of the manuscript is not affected by removing this part of the manuscript.

*3) Is there an estimate of how many axons are being recorded for trafficking events at the same time? This helps to understand the frequency of the movements better*.

Motor axons exit the VNC and in abdominal hemisegments A2–A7, project into the periphery along six nerves: the transverse nerve (TN), the intersegmental nerve (ISN), and four segmental nerve (SN) branches (from dorsal to ventral: SNa, SNb, SNc, and SNd) before innervating their specific target muscle(s) (75; 46). Several studies tried to analyse the number of axons of the 35–38 estimated motoneurons per hemisegment (46; 69; 63; 39) within each specific nerve. Our study mainly focused on the ISNs and SNs. By genetic mosaic techniques or single-cell dye labelling the number of axons within each nerve where determined to be 11–12 for ISN, 11 for ISNb/d, 4–7 for SNa, 11 SNb/d and 1 for SNc (30; 39).

4) What is the size of these ectopic puncta compared with synaptic BRP puncta at the active zone? How about the size of the moving puncta vs stationary puncta in the axon?

To address the first question we stained third instar wild type larva for BRP and measured the area [µm²] as well as the intensity of axonal and synaptic BRP punctae. As expected, axonal BRP punctae were on average about 50% smaller and less intense than synaptic BRP punctae (mean area of synaptic BRP^C-term^ punctae 0.1499 ± 0.0055 µm² in WT; mean area of axonal BRP^C-term^ punctae 0.08673 ± 0.0063 µm² in WT; mean intensity of synaptic BRP^C-term^ punctae 76.32 ± 6.027 AU in WT; mean intensity of axonal BRP^C-term^ punctae 43.98 ± 1.551 AU in WT n = 15 NMJs and 14 nerves; mean ± SEM; Figure 8).

Author response image 1.Comparison of synaptic vs. axonal BRP punctae(**A**) Quantification of synaptic vs axonal BRP punctae size. Mean area of synaptic BRP^C-term^ punctae 0.1499 ± 0.0055 µm² in WT; mean area of axonal BRP^C-term^ punctae 0.08673 ± 0.0063 µm² in WT; n = 15 NMJs and 14 nerves. (**B**) Quantification of synaptic vs axonal BRP punctae intensity. Mean intensity of synaptic BRP^C-term^ punctae 76.32 ± 6.02 AU in WT; mean intensity of axonal BRP^C-term^ punctae 43.98 ± 1.55 AU in WT; n = 15 NMJs and 14 nerves. All panels show mean values and errors bars representing SEM. *, p ≤ 0.05; **, p ≤ 0.01; ***, p ≤ 0.001; ns, not significant, p > 0.05, Mann–Whitney U test.**DOI:**
http://dx.doi.org/10.7554/eLife.06935.027

To address the second question concerning the size of the moving vs stationary axonal BRP punctae we quantified our existing intravital imaging data of *ctrl*, *aplip1*^*null*^, *aplip1*^*hypo*^ and *srpk79D*^*atc*^ and compared stationary vs moving punctae. Moving BRP punctae of ctrl (*Ok6::UAS-BRP*^*D3-straw*^) animals were significantly smaller than stationary punctae of all three mutant backgrounds (mean diameter of moving BRP^D3−straw^ punctae of *ctrl* 0.491 ± 0.008 nm; n = 20; mean diameter of stationary BRP^D3−straw^ punctae 1.066 ± 0.283 nm in *aplip1*^*null*^; n = 4; mean diameter of stationary BRP^D3−straw^ punctae 0.907 ± 0.110 nm in *aplip1*^*hypo*^; n = 12; mean diameter of stationary BRP^D3−straw^ punctae 1.363 ± 0.096 nm in *srpk79D*^*atc*^; n = 5; mean ± SEM; Figure 9).

Author response image 2.Comparison of moving vs. stationary axonal BRP punctae in *aplip1* and *srpk79D* mutant animals.(A) Quantification of mean diameter of moving vs stationary axonal BRP punctae. (Mean diameter of moving BRP^D3−straw^ punctae of *ctrl* 0.491 ± 0.008 nm; n = 20; mean diameter of stationary BRP^D3−straw^ punctae 1.066 ± 0.283 nm in *aplip1*^*null*^; n = 4; mean diameter of stationary BRP^D3−straw^ punctae 0.907 ± 0.110 nm in *aplip1*^*hypo*^; n = 12; mean diameter of stationary BRP^D3−straw^ punctae 1.363 ± 0.096 nm in *srpk79D*^*atc*^; n = 5; mean ± SEM). All panels show mean values and errors bars representing SEM. *, p ≤ 0.05; **, p ≤ 0.01; ***, p ≤ 0.001; ns, not significant, p > 0.05, one-way analysis of variance (ANOVA).**DOI:**
http://dx.doi.org/10.7554/eLife.06935.028